# Arpin deficiency increases actomyosin contractility and vascular permeability

Armando Montoya-Garcia[1], Idaira M Guerrero-Fonseca[1], Sandra D Chanez-Paredes[1†], Karina B Hernandez-Almaraz[1], Iliana I Leon-Vega[1], Angelica Silva-Olivares[2], Abigail Betanzos[2], Monica Mondragon-Castelan[3], Ricardo Mondragon-Flores[3], Citlaltepetl Salinas-Lara[4], Hilda Vargas-Robles[1], Michael Schnoor[1]*

[1]Department of Molecular Biomedicine, CINVESTAV-IPN, Mexico City, Mexico; [2]Department of Infectomics and Molecular Pathogenesis, CINVESTAV-IPN, Mexico City, Mexico; [3]Department of Biochemistry, CINVESTAV-IPN, Mexico City, Mexico; [4]Laboratorio de Patogénesis Molecular, Facultad de Estudios Superiores de Iztacala, Tlalnepantla de Baz, Mexico

*For correspondence:
mschnoor@cinvestav.mx

Present address: †Laboratory of Mucosal Barrier Pathobiology, Department of Pathology Brigham and Women's Hospital, Harvard Medical School, Boston, United States

Competing interest: The authors declare that no competing interests exist.

**Abstract** Arpin was discovered as an inhibitor of the Arp2/3 complex localized at the lamellipodial tip of fibroblasts, where it regulated migration steering. Recently, we showed that arpin stabilizes the epithelial barrier in an Arp2/3-dependent manner. However, the expression and functions of arpin in endothelial cells (EC) have not yet been described. Arpin mRNA and protein are expressed in EC and downregulated by pro-inflammatory cytokines. Arpin depletion in Human Umbilical Vein Endothelial Cells causes the formation of actomyosin stress fibers leading to increased permeability in an Arp2/3-independent manner. Instead, inhibitors of ROCK1 and ZIPK, kinases involved in the generation of stress fibers, normalize the loss-of-arpin effects on actin filaments and permeability. Arpin-deficient mice are viable but show a characteristic vascular phenotype in the lung including edema, microhemorrhage, and vascular congestion, increased F-actin levels, and vascular permeability. Our data show that, apart from being an Arp2/3 inhibitor, arpin is also a regulator of actomyosin contractility and endothelial barrier integrity.

## eLife assessment

This study presents **solid** results to demonstrate that arpin is expressed in the endothelium of blood vessels and that its deficiency leads to leaky blood vessels in in vivo and in vitro models. The work does not yet clarify the mechanistic connection between arpin and increased ROCK activity. The study adds some insights to our understanding of the complicated network of proteins that control this process, and it will be **useful** to individuals within this defined field of study.

## Introduction

Arpin was discovered 11 years ago in fibroblasts as a protein that binds to and inhibits the heptameric actin nucleator actin-related protein 2/3 (Arp2/3) complex (*Dang et al., 2013*). Arpin competed with WAVE2, a nucleation-promoting factor (NPF) that activates Arp2/3, at the lamellipodial tip of migrating fibroblasts to steer random migration by regulating migration speed and directional persistence (*Dang et al., 2013*). However, later studies demonstrated that arpin did not regulate chemotaxis (*Dang et al., 2017*). Subsequently, arpin has been studied in different cancer types (*Lomakina et al., 2016*; *Li et al., 2017b*; *Zhang et al., 2019a*), where its downregulation correlated with poor prognosis of the disease and high migratory capability and invasiveness of the tumor cells (*Li et al., 2017a*; *Lomakina*

et al., 2016). We have shown that arpin was expressed in colon epithelial cells and that patients with ulcerative colitis had lower levels of arpin in acutely inflamed tissue areas. Arpin colocalized with junctional proteins and stabilized epithelial barrier integrity under basal and inflammatory conditions in an Arp2/3-dependent manner (*Chánez-Paredes et al., 2021*). However, nothing is known about arpin expression in endothelial cells (EC) and its functions in endothelial barrier regulation during inflammation.

The endothelium forms a semi-permeable barrier that separates the blood from tissues. During inflammation, the weakening of cell-cell adhesive structures including adherens junctions (AJ) and tight junctions (TJ) is important for allowing the passage of plasma proteins and immune cells to the inflamed tissue and inflammation resolution (*Reglero-Real et al., 2016*). The correct regulation of AJ and TJ stability and thus vascular permeability is controlled by dynamic actin cytoskeletal remodeling including the formation of stabilizing cortical actin filaments and destabilizing contractile stress fibers (*Prasain and Stevens, 2009*; *García-Ponce et al., 2015*). As an important actin cytoskeletal regulator, the Arp2/3 complex regulates the maintenance and recovery of the pulmonary endothelial barrier (*Belvitch et al., 2017*). On the other hand, Arp2/3 is also involved in the endocytosis of occludin at the blood-brain barrier during inflammation (*Park et al., 2013*). Additionally, Arp2/3 and its activator WAVE2 stabilize endothelial barrier functions by promoting the formation of junction-associated inter-mittent lamellipodia (JAIL) (*Abu Taha et al., 2014*). These studies are only a few examples highlighting the importance of the Arp2/3 complex for controlling vascular permeability. While Arp2/3 activators such as WAVE2 have been studied in this context, the role of Arp2/3 inhibitors in endothelial barrier regulation remains elusive. Therefore, we hypothesized that arpin is expressed in endothelial cells and that it contributes to the regulation of actin cytoskeleton dynamics at junctions to fine-tune the regulation of endothelial permeability under basal and inflammatory conditions. To test this hypothesis, we generated arpin-depleted Human Umbilical Vein Endothelial Cells (HUVEC) and arpin-deficient mice to analyze the role of arpin in endothelial barrier regulation in vitro and in vivo.

## Results

### Arpin is expressed in endothelial cells

Arpin was discovered in fibroblasts (*Dang et al., 2013*), and it was subsequently described to be expressed in other cells including colon epithelial cells (*Chánez-Paredes et al., 2021*). However, it remains unknown whether arpin is expressed in endothelial cells. We found here by end-point RT-PCR that arpin mRNA is expressed in different human and mouse endothelial cells (*Figure 1A*). Also, arpin protein is present in HMEC-1 (Human Microvascular Endothelial Cells) and HUVEC at similar levels with the predicted molecular weight of 25 kDa (*Figure 1B*).

Arpin was localized throughout the cytosol in confluent HUVEC monolayers (*Figure 1C*). However, line scans revealed an enrichment of arpin close to vascular endothelial cadherin (VE-cadherin) at cell-cell contacts. (*Figure 1C*). Analyzing arpin localization in postcapillary venules of cremaster muscles using Platelet-Endothelial Cell Adhesion Molecule-1 (PECAM-1) as the template for endothelial cell-cell contacts, we observed a similar localization of arpin throughout the endothelial cytosol with an enrichment near and at junctions (*Figure 1D*). Although the majority of arpin is localized in the cytosol, such enrichment at junctions observed both in human HUVEC culture and mouse cremaster venules suggests that arpin could contribute to endothelial barrier regulation.

### Arpin is downregulated by inflammatory cytokines

Endothelial cells are activated by pro-inflammatory cytokines such as TNFα and IL-1β leading to weakening of the endothelial barrier (*Chen et al., 2021*; *Makó et al., 2010*). To study arpin regulation during inflammation, we analyzed *ARPIN* gene expression by qRT-PCR in HUVEC monolayers treated with TNFα for 4 and 18 hr and found a significant downregulation of around 60% (*Figure 2A*). Remarkably, the expression of the other Arp2/3 inhibitors, *AP1AR* and *PICK1*, was not significantly changed after TNFα treatment. Western blot and densitometry analysis confirmed this decrease of arpin at the protein level (*Figure 2B*) with no significant changes in PICK1 protein levels (*Figure 2—figure supplement 1A*). Upregulation of Intercellular Adhesion Molecule-1 (ICAM-1) was used here as a positive control for the inflammatory response.

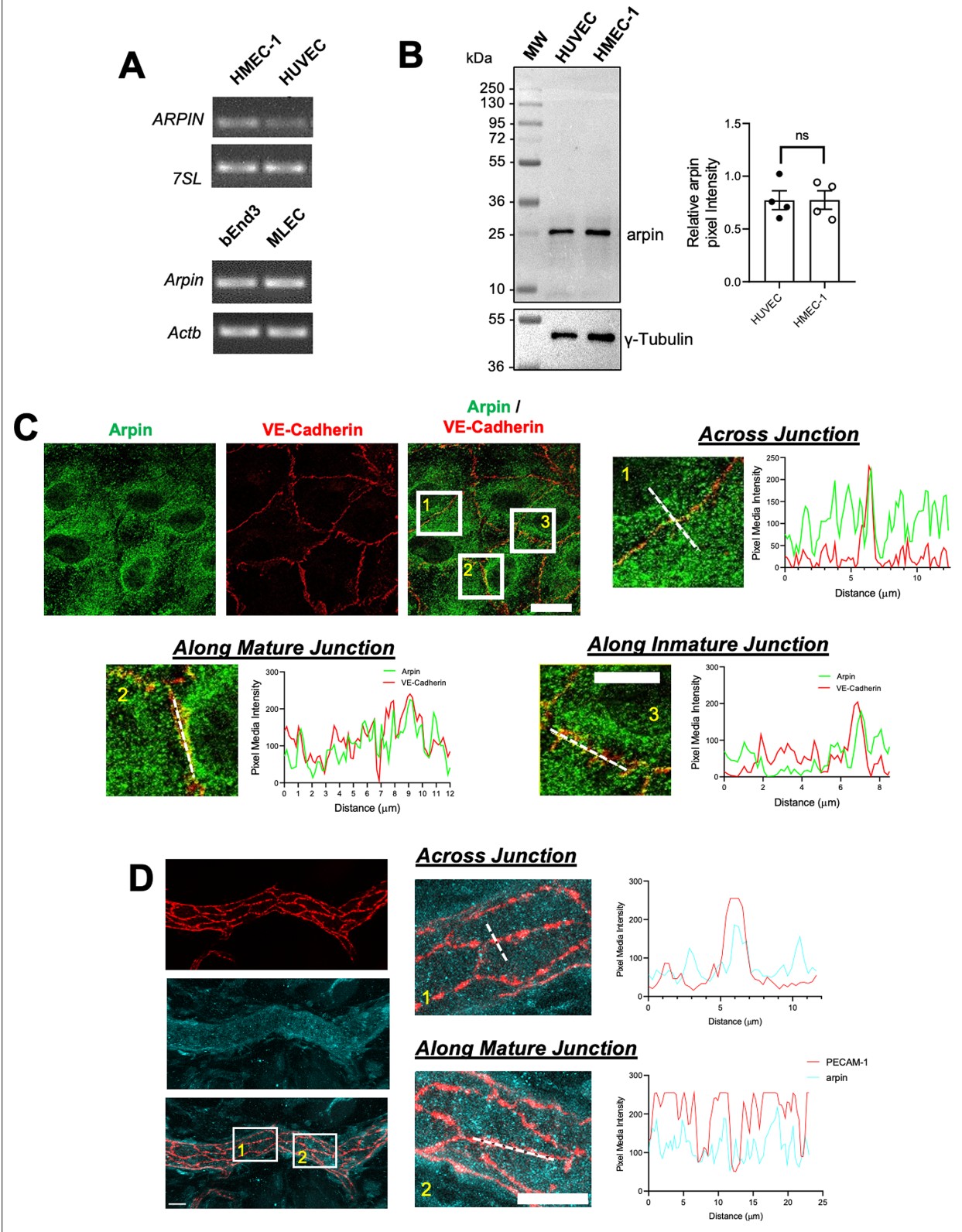

**Figure 1.** Arpin is expressed in endothelial cells. (**A**) End-point RT-PCR for *ARPIN* in the human microvascular endothelial cell line (HMEC-1) and human umbilical vein endothelial cells (HUVEC), *7SL* was used as housekeeping gene (top); and *Arpin* in the mouse brain endothelial cell line bEnd.3 and primary murine lung endothelial cells (MLEC); *Actb* was used as a housekeeping gene (bottom). A representative gel of three independent experiments is shown. (**B**) Representative western blot for arpin in HUVEC and HMEC-1. MW = molecular weight bands. The graph shows the quantification of the

*Figure 1 continued on next page*

*Figure 1 continued*

relative pixel intensity of the arpin band normalized to γ-tubulin as a loading control (n=4). Data are represented as mean ± standard error of the mean (SEM); ns: non-significant; two-tailed student's t-test. (**C**) Representative immunostaining of arpin and vascular endothelial (VE)-Cadherin in HUVEC to analyze arpin localization at cell-cell junctions (40 X objective; scale bar = 20 μm). Magnified views of boxed regions are shown (4.3 digital zooms; scale bar = 5 μm). The graphs show the dashed line scans performed with ImageJ software along the white lines in the magnified images. Profiles of the pixel intensity mean of arpin and VE-Cadherin across junctions (1), along mature junctions (2), and along immature junctions (3) are depicted. 25 images each were analyzed from three independent experiments. (**D**) Representative immunostaining of arpin and PECAM-1 in postcapillary venules of mouse cremaster muscles to analyze arpin localization at cell-cell junctions in vivo (40 X objective; scale bar = 20 μm). Magnified views of dashed boxed regions are shown (3.3 X digital zoom; scale bar = 10 μm). The graphs show the line scans performed with ImageJ software along the white lines in the magnified images. Profiles of the pixel intensity mean of arpin and PECAM-1 across junctions (1), and along mature junctions (2) are depicted. 15 venules each were analyzed from four mice.

The online version of this article includes the following source data for figure 1:

**Source data 1.** Uncropped and labeled gels and membranes for *Figure 1*.

**Source data 2.** Raw unedited membranes and gels for *Figure 1*.

Immunofluorescence of arpin in HUVEC monolayers treated or not with TNFα confirmed downregulation of arpin (*Figure 2C*). This decrease was not only obvious for total arpin, but arpin was also lost at junctions (arrows in the magnifications of *Figure 2C*) and actin filaments (Top-right graphs *Figure 2C*). Interestingly, arpin reduction significantly correlated with the TNFα-induced increase in actin stress fibers (Below graph *Figure 2C*; *Wójciak-Stothard et al., 1998*). Arpin protein downregulation was also observed after IL-1β treatment in HUVECs for 4 and 18 h (*Figure 2—figure supplement 1B*) and no significant changes were observed in the protein levels of PICK1 (*Figure 2—figure supplement 1C*). Similar to TNFα-treatment, IL-1β induces loss of arpin at all subcellular locations (top-right graphs *Figure 2—figure supplement 1D*). Again, IL1β-induced arpin downregulation significantly correlated with an increase in actin stress fibers (lower graph *Figure 2—figure supplement 1D*).

Arpin immunofluorescence signal in post-capillary venules of the TNFα-inflamed cremaster muscle was also reduced by around 50%, which was again true for both junctional and non-junctional arpin (*Figure 2D*). Robust neutrophil recruitment confirmed that inflammation was induced correctly in this model. Overall, endothelial arpin is downregulated in response to pro-inflammatory cytokines in vitro and in vivo suggesting that arpin plays an important role in the regulation of endothelial barrier dysfunction during inflammation.

## Arpin depletion increases endothelial permeability and modifies junction architecture

To analyze the function of arpin in endothelial barrier regulation, we generated arpin-depleted HUVEC with more than 80% reduction in arpin protein levels (*Figure 3A*). Of note, arpin depletion did not change the expression of arpC5, a subunit of the Arp2/3 complex, or the NPFs WAVE2 and WASP (*Figure 3A*). Although arpin-depleted HUVEC formed monolayers, the paracellular flux of 150 kDa FITC-dextran across untreated arpin-depleted HUVEC monolayers was 34.5% (±3.70) higher (*Figure 3B*). While TNFα treatment of control HUVEC induced the expected strong increase in permeability, TNFα treatment of arpin-depleted HUVEC showed an even higher increase compared to control HUVEC treated with TNFα (*Figure 3B*) suggesting that the TNFα-induced and arpin-induced permeability increases are regulated by different mechanisms. However, we did not observe changes in the total protein levels of the AJ proteins VE-Cadherin, β-catenin, and vinculin in control and arpin-depleted HUVEC (*Figure 3C*), or in the levels of the TJ proteins ZO-1 and claudin-5 (*Figure 3D*). However, immunofluorescence staining of VE-Cadherin and β-catenin revealed discontinuous and zipper-like junctional patterns, indicative of immature junctions in arpin-depleted HUVEC (*Figure 3E*). Surprisingly, the calcium switch assay showed that control and arpin-depleted HUVEC formed new junctions in a similar fashion (*Figure 3—figure supplement 1A, B*). These data suggest that arpin is not required for junction assembly, but rather for stabilizing existing junctions.

## Arpin depletion induces actin filament formation

It is well known that the endothelial actin cytoskeleton contributes to barrier homeostasis and that significant actin cytoskeleton remodeling accompanies increased vascular permeability (*García Ponce et al., 2016*; *Hilfenhaus et al., 2018*; *Waschke et al., 2005*). Thus, we analyzed F-actin using

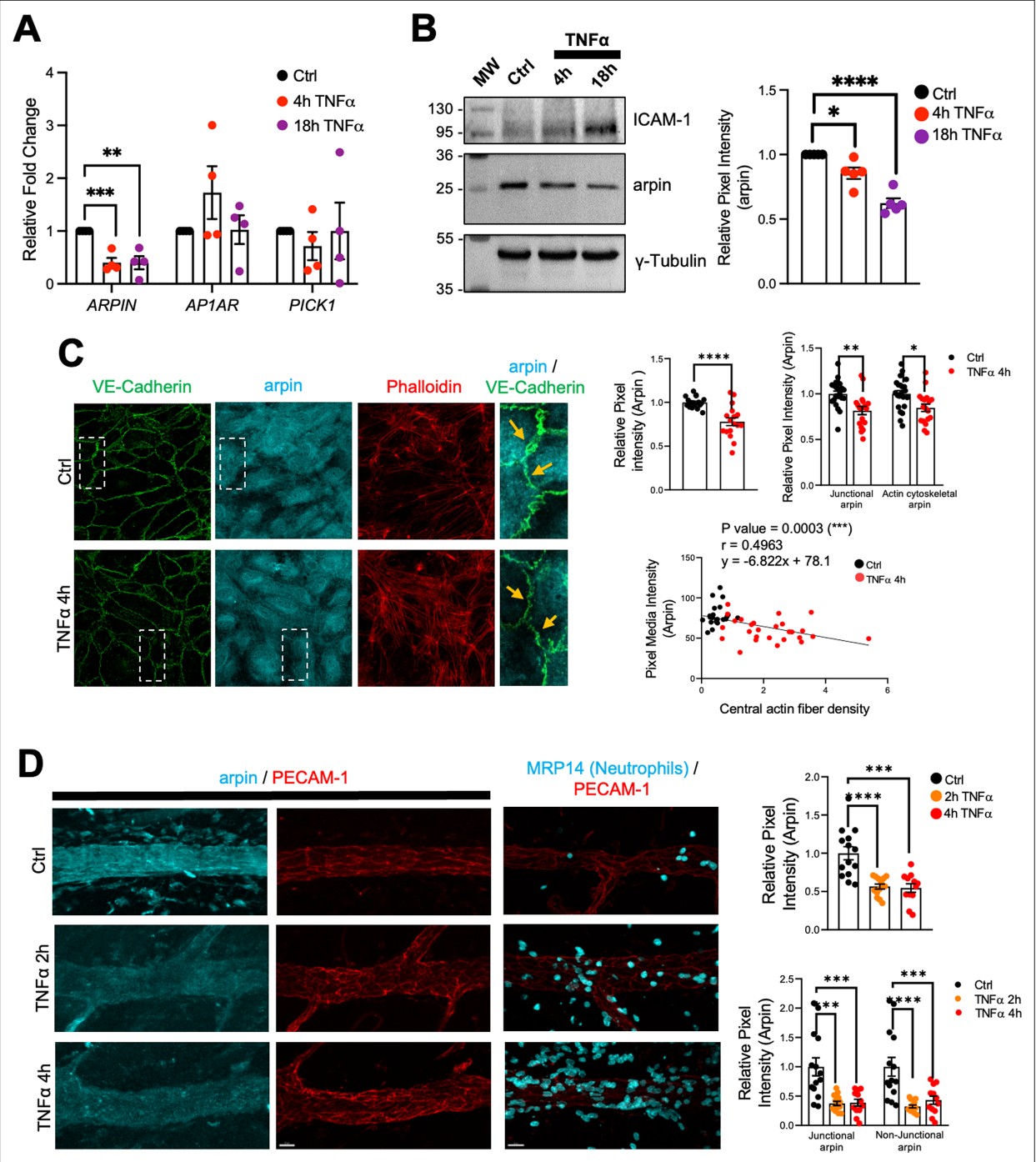

**Figure 2.** Arpin is downregulated by TNFα. (**A**) qRT-PCR for the Arp2/3 complex inhibitors *ARPIN*, *AP1AR,* and *PICK1* using cDNA from human umbilical vein endothelial cells (HUVEC) treated or not with TNFα for the indicated times (n=4). Data are shown as relative expression (fold change) normalized to the housekeeping gene *7SL*. (**B**) Representative western blot for arpin in HUVEC treated or not with TNFα for the indicated times (n=5). MW = molecular weight bands. ICAM-1 was used as a positive control of the induction of inflammation. The graph shows the mean pixel intensity of arpin bands normalized to tubulin as loading control and to the control condition (set to 1). (**C**) Representative immunostaining of arpin and vascular endothelial (VE)-Cadherin together with phalloidin F-actin staining in HUVEC treated or not with TNFα for 4 hr (40 X Objective, scale bar = 20 μm). Magnifications of the right show with the orange arrows loss of arpin at junctions with TNFα treatment. The graphs show arpin pixel intensity quantification after treatment normalized to the average of ctrl HUVECs: top-left, total arpin in 17 images each was analyzed from four independent experiments; top-right, junctional, and actin cytoskeletal arpin in 19 images each were analyzed from three independent experiments; and bottom graph: Pearson's correlation analysis showing that arpin is inversely correlated to central actin fiber density (Pearson's correlation coefficient, r; 49 cells were analyzed from three independent experiments). (**D**) Representative immunostaining for arpin (blue) and PECAM-1 (red) in postcapillary venules

*Figure 2 continued on next page*

*Figure 2 continued*

treated or not with TNFα at the indicated times. PECAM-1 and MRP14 (neutrophils) stainings (right) are shown as positive controls for the induction of inflammation (40 x objective, scale bar = 20 μm). The top graph shows total arpin pixel intensity quantification after TNFα treatment normalized to the average of control venules (12–14 venules were analyzed from four mice in each group). The bottom graph shows junctional and non-junctional arpin quantification after TNFα treatment normalized to the average of control venules (12–14 venules were analyzed from four mice in each group). All data are represented as mean ± SEM; *p<0.05; **p<0.01; ***p<0.001; ****p<0.0001; two-tailed student's t-test.

The online version of this article includes the following source data and figure supplement(s) for figure 2:

**Source data 1.** Uncropped and labeled membranes for *Figure 2*.

**Source data 2.** Raw unedited membranes for *Figure 2*.

**Figure supplement 1.** Arpin is reduced by IL-1β.

**Figure supplement 1—source data 1.** Uncropped and labeled membranes for *Figure 2—figure supplement 1*.

**Figure supplement 1—source data 2.** Raw unedited membranes for *Figure 2—figure supplement 1*.

phalloidin staining in control and arpin-depleted HUVEC. Quantifying the pixel intensity of the phalloidin stainings, we observed a statistically significant increase in actin filaments in the cells with arpin deficiency (*Figure 4A*). Particularly, arpin-depleted HUVEC showed a significant increase in central actin filaments crossing the cell body (*Figure 4B*). However, we did not observe significant differences in the total actin protein levels in control and arpin-depleted HUVEC (*Figure 4C*) suggesting that G-actin levels are reduced at the expense of F-actin formation. Quantification of other proteins known to be involved in actin dynamics such as cortactin, coronin1B, and cofilin did not reveal any significant changes caused by the loss of arpin (*Figure 4D*). In the next set of experiments, we wanted to identify the nature of these actin filaments and answer the question of whether the observed hyperpermeability is related to these changes in the actin cytoskeleton.

## Inhibition of the Arp2/3 complex does not rescue the observed phenotype in arpin-depleted HUVEC

The first function of arpin described was that it inhibits the Arp2/3 complex (*Dang et al., 2013*). Assuming this function, loss of arpin would mean more active Arp2/3. To determine if a potential increase in Arp2/3 activity is causing the observed effects in arpin-depleted endothelial cells, we used the pharmacological Arp2/3 inhibitor CK666 to test whether it would reverse the observed phenotype (*Figure 5*). Surprisingly, phalloidin stainings and quantification of the total F-actin signal revealed no reduction in the pixel intensity between CK666-treated and vehicle-treated arpin-depleted HUVEC (*Figure 5A and B*). The central actin fiber density was also not affected by Arp2/3 inhibition in arpin-depleted HUVEC (*Figure 5C*). Interestingly, permeability increased in control HUVEC after CK666 treatment (*Figure 5D*). Instead of normalizing the permeability in arpin-depleted HUVEC, CK666 induced an additional increase in permeability (*Figure 5D*). These surprising results demonstrate that arpin regulates actin filament formation and permeability in HUVEC in an Arp2/3-independent manner.

## Arpin depletion leads to increased formation of actin stress fibers

Given that arpin acts independently of the Arp2/3 complex in HUVEC and that the observed central actin filaments in arpin-depleted HUVECs resemble stress fibers (*Tojkander et al., 2012*), we investigated whether these actin structures are contractile actomyosin stress fibers, which are known to exert mechanical forces on junctions and thus destabilize endothelial contacts (*Millán et al., 2010*). As shown in *Figure 6A*, we observed a significant increase in the phosphorylation of myosin light chain (MLC) at serine-19 in arpin-depleted HUVECs, indicative of actomyosin contractility. MLC is a target of kinases such as ROCK1 and ZIPK that phosphorylate it, and phosphatases such as Myosin Phosphatase Target Subunit 1 (MYPT1) that dephosphorylate MLC. Concomitant with the increase in p-MLC-S19, we found an increase in the phosphorylation of MYPT1 at Threonin-696 and Threonin-853 in arpin-depleted HUVECs (*Figure 6B*), phosphorylations known to inactivate MYPT1. In agreement, we observed increased total levels of ROCK1 and ZIPK in arpin-depleted HUVECs that likely contribute to increased MLC phosphorylation (*Figure 6C*). However, the levels of the formin mDia, implicated in the nucleation of actin stress fibers, were not affected by arpin depletion (*Figure 6C*). These results suggest that the increased central actin filaments in arpin-depleted HUVEC are contractile actomyosin stress fibers that contribute to the destabilization of endothelial junctions and increased permeability.

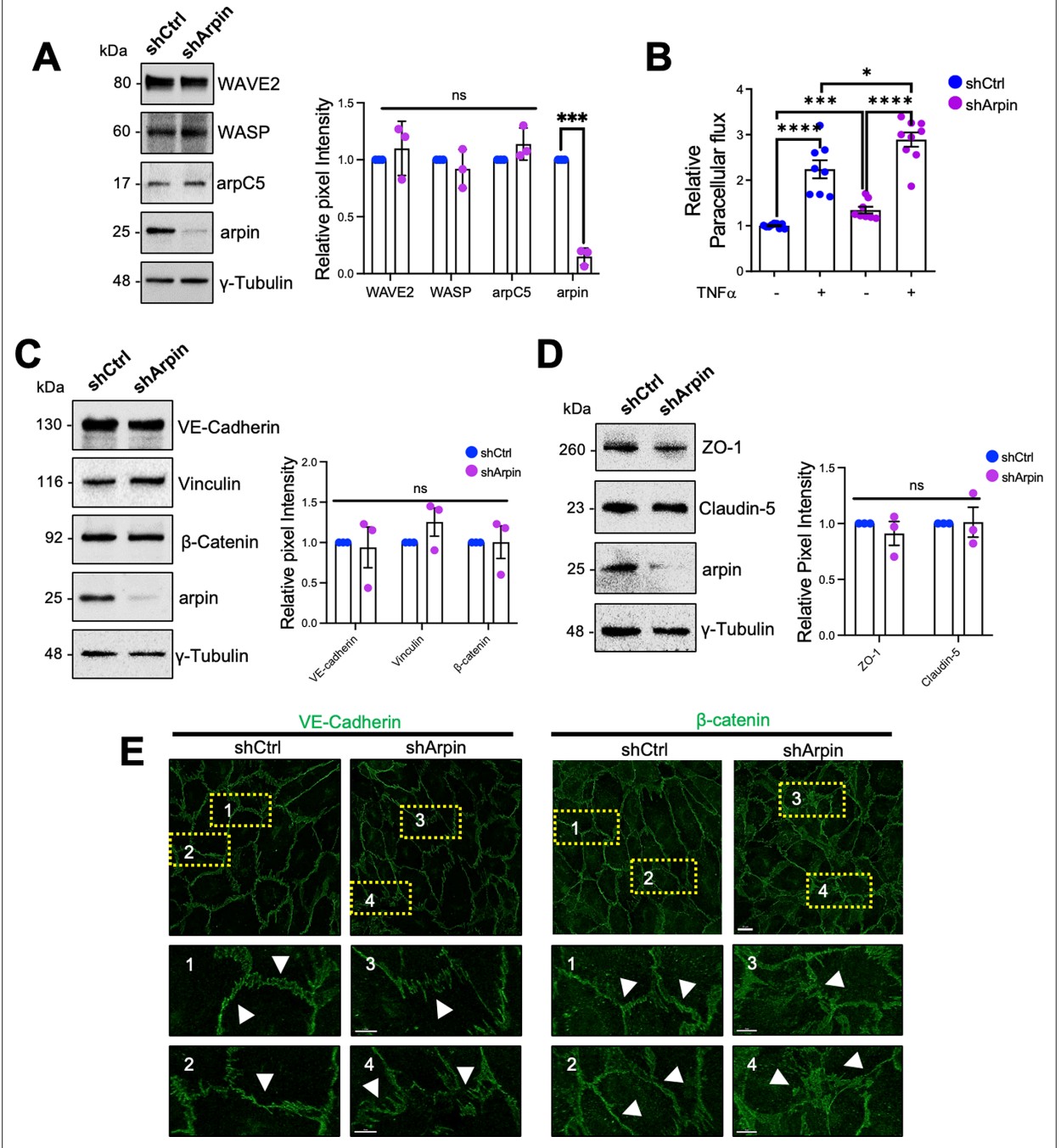

**Figure 3.** Arpin depletion induces hyperpermeability and irregular junction patterns. (**A**) Representative Western blots for WASP family Verprolin homolog 2 (WAVE2), WASP, the Arp2/3 subunit ArpC5, and arpin in lysates of control (shCtrl) and arpin-depleted (shArpin) human umbilical vein endothelial cells (HUVEC). The graph shows the mean pixel intensities of the protein bands normalized to the respective control bands. All bands were normalized to γ-tubulin as loading control (n=3). (**B**) Paracellular flux of 150 kDa FITC-dextran across confluent control and arpin-depleted HUVEC monolayers cultured on transwell filters (0.4 μm pore size) untreated or treated with TNFα for 18 hr. Data are represented as relative permeability normalized to control HUVEC set to 1 (n=9 in three independent experiments). (**C**) Representative Western blots for vascular endothelial (VE)-Cadherin, β-catenin, vinculin, and arpin in lysates of control and arpin-depleted HUVEC. The graph shows the mean pixel intensities of the protein bands normalized to the respective control bands (set to 1). All bands were normalized to γ-tubulin as a loading control (n=3). (**D**) Representative Western blots for ZO-1, claudin-5, and arpin in lysates of control and arpin-depleted HUVEC. γ-Tubulin was used as a loading control. The graph shows the quantification of the relative pixel intensity of ZO-1 and claudin-5 bands normalized to the untreated control (set to 1) and γ-tubulin (n=3). (**E**) Immunostaining of VE-Cadherin and β-catenin in control and arpin-depleted HUVEC (40 X objective, scale bar = 20 μm; dashed boxes indicate magnified areas that highlight changes in junctional architecture; 3.3 digital zoom, scale bars = 5 μm). Representative images of four independent

*Figure 3 continued*

experiments are shown. Arrows depict linear and mature junctions in shctrl HUVEC and interrupted and gap junction formation and junction internalization in shArpin HUVEC. All data are represented as mean ± SEM; ns: non-significant; *p<0.05; ***p<0.001; ****p<0.0001; two-tailed student's t-test.

The online version of this article includes the following source data and figure supplement(s) for figure 3:

**Source data 1.** Uncropped and labeled membranes for *Figure 3*.

**Source data 2.** Raw unedited membranes for *Figure 3*.

**Figure supplement 1.** Arpin does not participate in junction assembly.

## Inhibition of ROCK1/2 and ZIPK rescues endothelial barrier functions in arpin-depleted HUVEC

Given the increased levels of ROCK1 and ZIPK and the increased phosphorylation of MLC and MYPT1, we hypothesized that pharmacological inhibition of ROCK1 by Y27632 and of ZIPK by HS38 could revert increased stress fiber formation and permeability in arpin-depleted HUVEC. Phalloidin staining in arpin-depleted HUVECs treated with Y27632 (*Figure 7A*) showed indeed a reduction of the total levels of actin filaments (*Figure 7B*) and the central actin fiber density (*Figure 7C*) when compared to vehicle-treated arpin-depleted HUVEC. However, it was not reduced to the levels seen in Y27632-treated control HUVEC that were significantly lower compared to vehicle-treated control cells. Similar results were observed in permeability assays (*Figure 7D*). While the permeability in Y27632-treated arpin-depleted HUVEC were reduced to the levels of vehicle-treated control HUVEC, they were still higher compared to Y27632-treated control HUVEC.

Moreover, we observed that arpin-depleted HUVECs treated with the ZIPK inhibitor HS38 showed total actin levels (*Figure 7E and F*) and central actin fiber density (*Figure 7G*) similar to control HUVEC treated with HS38 or vehicle. In contrast to Y27632, HS38 was not able to reduce actin filament formation in control HUVEC. Permeability assays confirmed these data and showed a complete rescue in arpin-depleted HUVEC treated with HS38 (*Figure 7H*). Of note, the even higher permeability in TNFα-treated arpin-depleted HUVECs was reduced after HS38 treatment but not to the levels of control cells indicating that ZIPK inhibition is able to rescue the permeability effect caused by the loss of arpin, but not the combined effect of TNFα treatment and loss of arpin (*Figure 7I*). This result further suggests that TNFα-induced permeability follows a different mechanism than the permeability effect caused by arpin depletion.

## Generation and characterization of arpin-deficient mice

Finally, to unravel the role of arpin in vascular permeability, we generated an arpin knockout mouse model on the C57BL/6 genetic background using CRISPR/Cas9-mediated genome engineering targeting exon 3 with a pair of specific gRNA sequences (*Figure 8A*). PCR of genomic DNA samples extracted from the tails clearly distinguishes the three genotypes (*Figure 8B*), and Western blot of arpin revealed its presence in different organs of wild-type mice, and its absence in knock-out mice (*Figure 8C*). $Arpin^{-/-}$ mice did not show any obvious phenotype and could not be distinguished from $Arpin^{+/+}$ mice by visual inspection. We did not observe changes in development, appearance, weight (*Figure 8—figure supplement 1A*), or susceptibility to diseases compared to $Arpin^{+/+}$ mice in pathogen-free conditions. Breeding female and male $Arpin^{+/-}$ mice generated offspring according to Mendelian ratios (*Supplementary file 1a*).

## Arpin deficiency causes increased vascular permeability

To unravel whether arpin also regulates endothelial barrier integrity in vivo, we performed permeability assays in vivo in the lung because $Arpin^{+/+}$ mice showed high levels of arpin in the lung (*Figure 8C*) and the lung is a highly vascularized organ. Remarkably, permeability in the lungs of $Arpin^{-/-}$ mice was around two-fold higher than in $Arpin^{+/+}$ mice under basal conditions without any permeability-inducing stimulus (*Figure 8D*). To rule out that this is an effect specific only for the lung, we also performed Miles assays to measure permeability in the back skin under basal conditions and with histamine as a permeability-inducing stimulus. Also in the skin, vascular permeability was higher in $Arpin^{-/-}$ mice under basal conditions compared to $Arpin^{+/+}$ mice (*Figure 8E*). Of note, histamine stimulation further induced permeability in $Arpin^{-/-}$ mice. Treatment with either the ROCK1/2 inhibitor

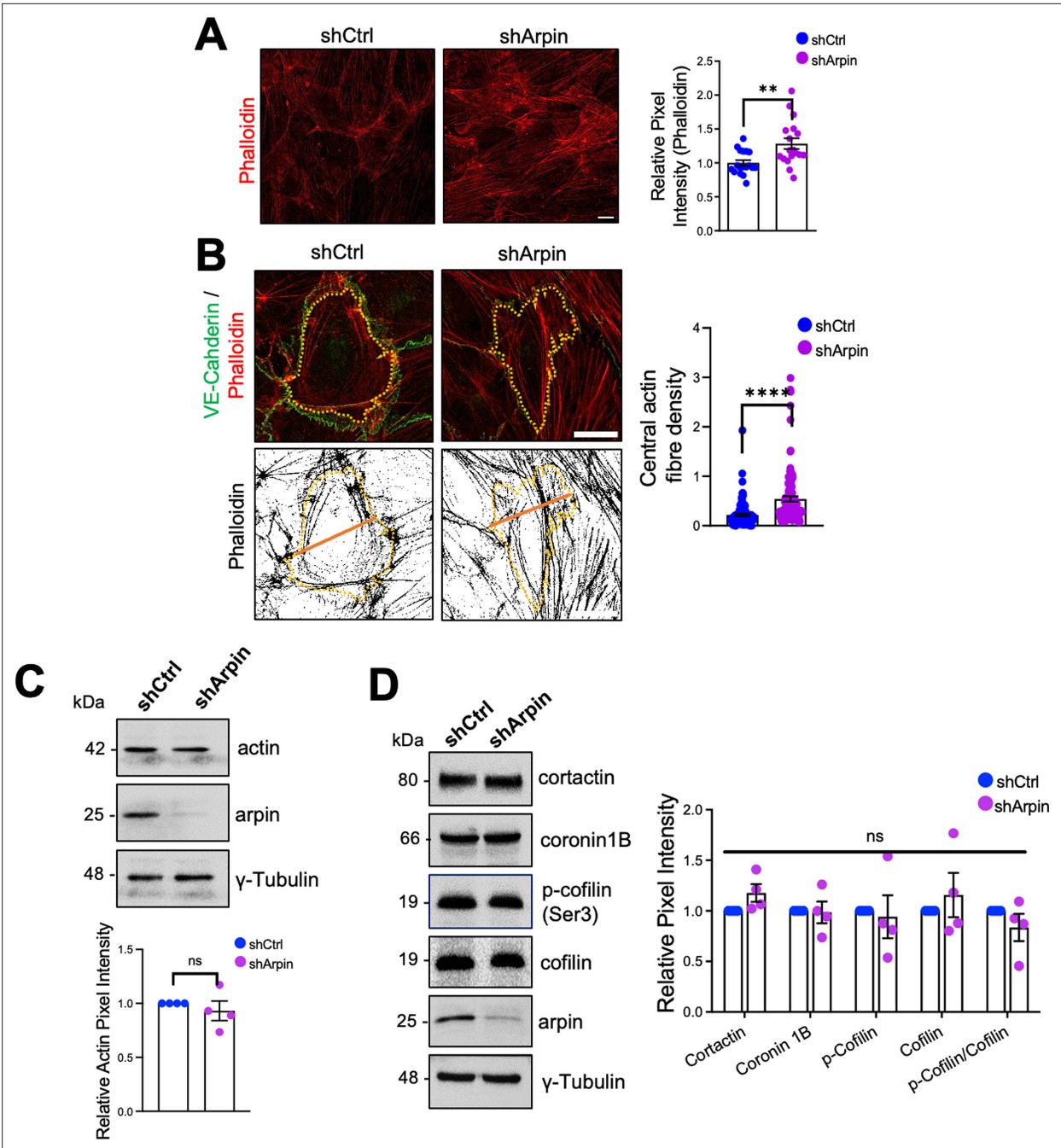

**Figure 4.** Arpin depletion induces actin filament formation. (**A**) Representative F-actin staining using phalloidin of control and arpin-depleted human umbilical vein endothelial cells (HUVEC) (40 X objective, scale bar = 20 μm). The graph shows phalloidin pixel intensity quantification normalized to the average of control HUVEC. 18 images were analyzed in each group from four independent experiments. (**B**) Representative immunostaining of vascular endothelial (VE)-Cadherin with F-actin staining using phalloidin (top) in control and arpin-depleted HUVEC. The method of central actin fiber quantification using the phalloidin signal is depicted on the bottom as reported before (***García Ponce et al., 2016***) (63 X objective, scale bar = 5 μm). Yellow dashed lines delineate an individual cell. The graph shows the mean central actin fiber density quantification along the orange lines. 109 cells were analyzed in each group from three independent experiments. (**C**) Representative Western blot for total actin in lysates of control and arpin-depleted HUVEC. The graph shows the mean pixel intensity of the actin bands normalized to the control and to γ-tubulin as a loading control (n=4). (**D**) Representative Western blots for cortactin, coronin1B, p-cofilin, cofilin, and arpin in lysates of control and arpin-depleted HUVEC. The graph shows the mean pixel intensity of the protein bands normalized to the respective control bands (set to 1). All bands were normalized to γ-tubulin as a loading control (n=4). All data are represented as mean ± SEM; ns: non-significant; **p<0.01; ****p<0.0001; two-tailed student's t-test.

The online version of this article includes the following source data for figure 4:

*Figure 4 continued on next page*

*Figure 4 continued*

**Source data 1.** Uncropped and labeled membranes for *Figure 4*.

**Source data 2.** Raw unedited membranes for *Figure 4*.

Y27632 or the ZIPK inhibitor HS38 reverted permeability to the basal levels observed in *Arpin*[+/+] mice, thus confirming our findings in arpin-depleted HUVEC. Interestingly, Arp2/3 inhibition using CK-666 did not affect vascular permeability in the skin of *Arpin*[-/-] mice (*Figure 8E*). Together, these results support the idea that arpin controls endothelial permeability via ROCK/ZIPK-dependent stress fiber formation and not via inhibition of the Arp2/3 complex. Next, we analyzed by Western blot overall expression levels of different junction proteins but did not observe any significant differences in the protein levels of VE-Cadherin, β-catenin, vinculin, claudin-1 and –5, and tubulin in *Arpin*[+/+] and *Arpin*[-/-] lungs (*Figure 8—figure supplement 1B*).

Histological analyses revealed normal lung histology in *Arpin*[+/+] mice (*Figure 8F*, Images 1 and 2). By contrast, the *Arpin*[-/-] lungs showed several abnormalities indicative of a vascular phenotype. For example, the absence of arpin caused a reduction of the volume of the alveolar space (*Figure 8F*, Image 3, asterisk), signs of hemorrhage and microhemorrhage (*Figure 8F*, Images 4 and 5, arrow-heads), and the presence of interalveolar edema (*Figure 8F*, Image 6, asterisks). Accordingly, the histological score was significantly increased in *Arpin*[-/-] mice (*Figure 8F*, graph below). Lungs of *Arpin*[-/-] mice also showed the presence of lymphocytes and macrophages. However, hemograms revealed that the number of leukocytes in the peripheral blood was not significantly altered in *Arpin*[-/-] mice

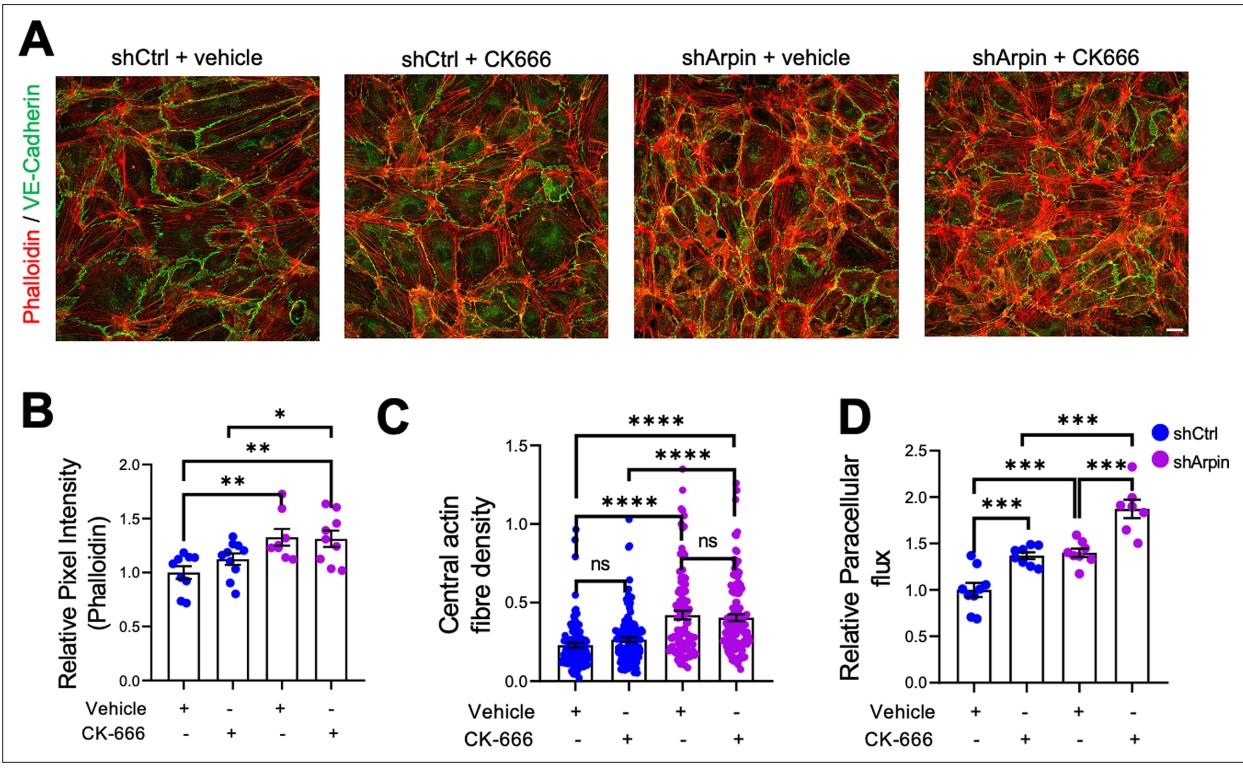

**Figure 5.** Inhibition of the Arp2/3 complex in arpin-depleted human umbilical vein endothelial cells (HUVEC) does not rescue hyperpermeability and actin cytoskeleton alterations. (**A**) Immunostaining of vascular endothelial (VE)-Cadherin with F-actin staining using phalloidin in control and arpin-depleted HUVEC treated with 100 µM of the Arp2/3 inhibitor CK-666 or vehicle (40 X objective, scale bar = 20 µm). Representative images of three independent experiments are shown. (**B**) Total phalloidin pixel intensity quantification normalized to the average of control HUVEC. Nine images were analyzed in each group from three independent experiments. (**C**) Central actin fiber density quantification done as described in *Figure 4B*. 100 cells were analyzed in each group from three independent experiments. (**D**) Paracellular flux assays using confluent control and arpin-depleted HUVEC monolayers on 0.4 µm pore transwell filters treated with 100 µM CK-666 or vehicle (n=7–9 from three independent experiments). Data are normalized to control HUVEC treated with the vehicle set to 1. All data are represented as mean ± SEM; ns: non-significant; *p<0.05; **p<0.01; ***p<0.001; ****p<0.0001; two-tailed student's t-test.

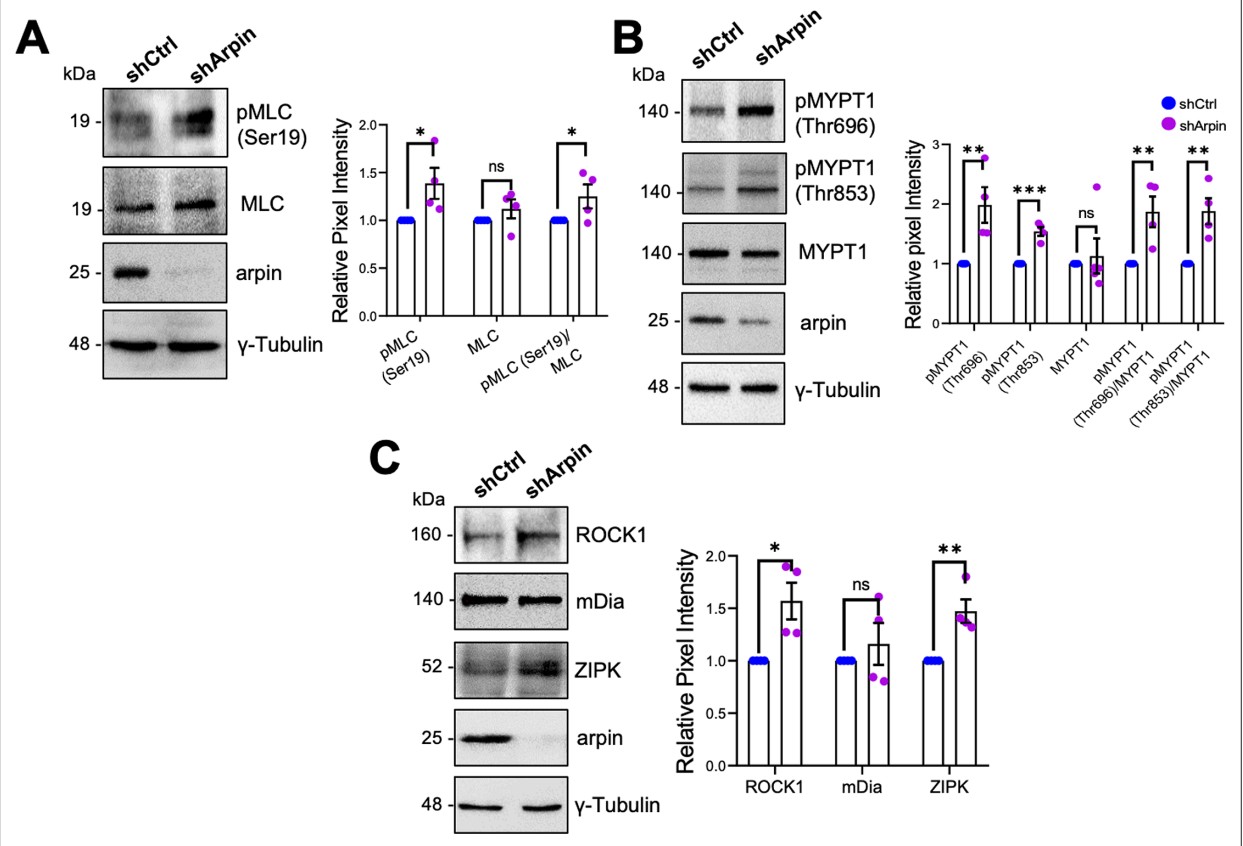

**Figure 6.** Arpin depletion increases the formation of actin stress fibers. (**A**) Representative Western blots for pMLC (Ser19), myosin light chain (MLC), and arpin in lysates of control and arpin-depleted human umbilical vein endothelial cells (HUVEC). The graph shows the mean pixel intensity of the protein bands normalized to the respective control band (set to 1). All bands were normalized to γ-tubulin as a loading control (n=4). (**B**) Representative Western blots for pMYPT1 (Thr696), pMYPT1 (Thr853), MYPT1, and arpin in lysates of control and arpin-depleted HUVEC. The graph shows the mean pixel intensity of the protein bands normalized to the respective control band (set to 1). All bands were normalized to γ-tubulin as a loading control (n=4). (**C**) Representative Western blots for Rho-associated protein kinase 1 (ROCK1), mDia, Zipper interacting protein kinase (ZIPK), and arpin in lysates of control and arpin-depleted HUVEC. The graph shows the mean pixel intensity of the protein bands normalized to the respective control band (set to 1). All bands were normalized to γ-tubulin as a loading control (n=4). All data are represented as mean ± SEM; ns: non-significant; *p<0.05; **p<0.01; ***p<0.001; two-tailed student's t-test.

The online version of this article includes the following source data for figure 6:

**Source data 1.** Uncropped and labeled membranes for *Figure 6*.

**Source data 2.** Raw unedited membranes for *Figure 6*.

compared with *Arpin*[+/+] mice (***Supplementary file 1b***), indicating that the presence of lymphocytes and macrophages in the lungs of *Arpin*[-/-] mice is not simply due to an increased number of these cells.

PECAM-1 immunostaining was performed to visualize blood vessels in the lung, and phalloidin to identify F-actin (***Figure 8G***). Although we did not observe differences in total levels of PECAM-1 in *Arpin*[+/+] and *Arpin*[-/-] lungs, we observed a significant increase in the total levels of F-actin (***Figure 8G***, graph below), similar to what we observed in arpin-depleted HUVEC suggesting that the increased permeability in vivo is also due to an increase in the formation of actin stress fibers. In addition, we also analyzed endothelial junctions in post-capillary venules of cremaster muscles under basal conditions. We observed in venules of *Arpin*[-/-] mice VE-Cadherin internalization leading to a zipper-like junction structure similar to what we observed in arpin-depleted HUVEC (***Figure 8—figure supplement 1C***). Also, ZO-1 staining showed gaps in *Arpin*[-/-] mice, whereas it was continuous in *Arpin*[+/+] mice (***Figure 8—figure supplement 1D***). Overall, our data highlight a novel important Arp2/3-independent role of arpin in maintaining endothelial barrier integrity in vitro and in vivo.

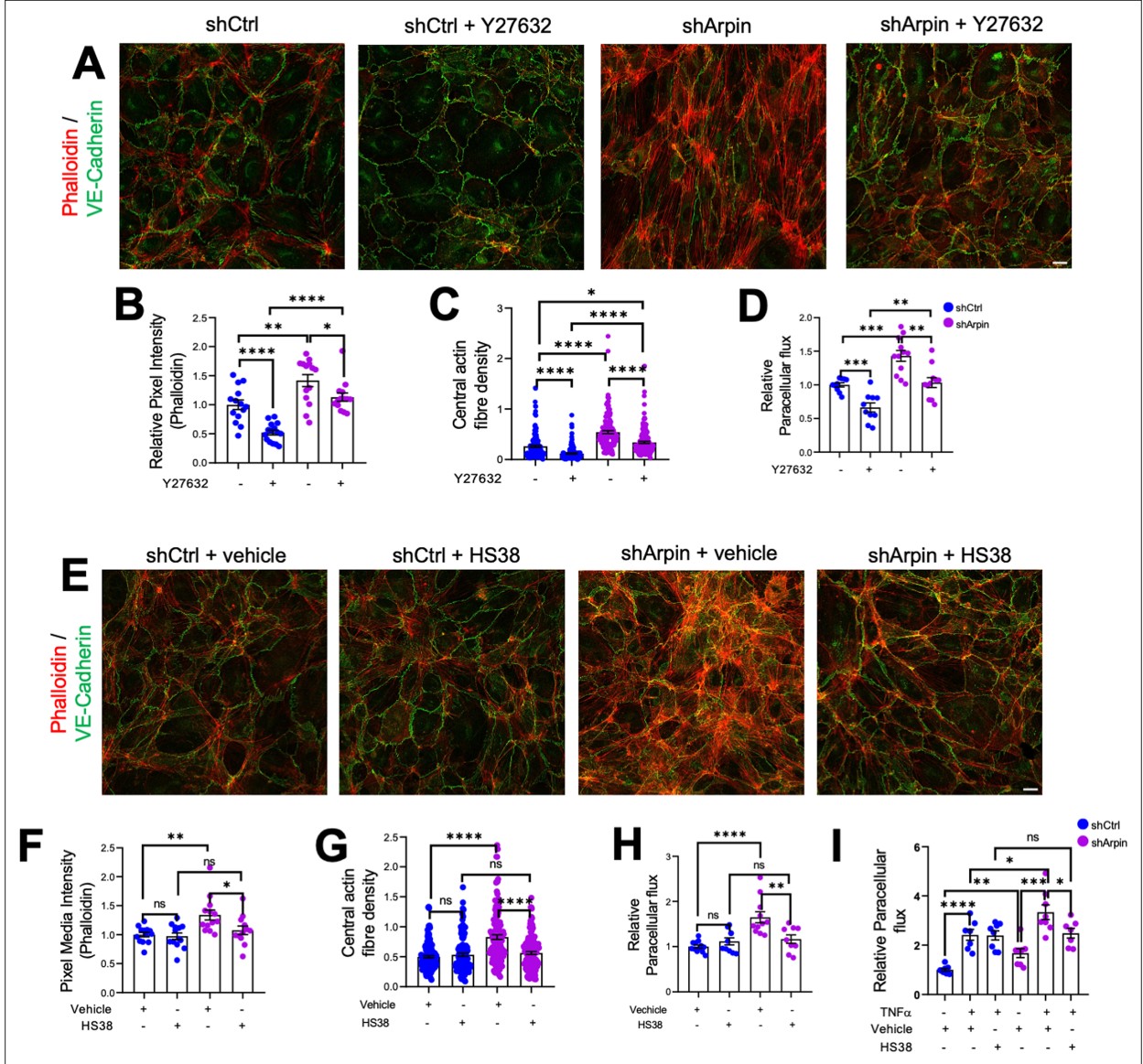

**Figure 7.** ROCK1/2 and Zipper interacting protein kinase (ZIPK) inhibition partially rescues the increase in stress fibers and permeability in arpin-depleted human umbilical vein endothelial cells (HUVEC). (**A**) Representative immunostaining of vascular endothelial (VE)-Cadherin with F-actin staining using phalloidin in control and arpin-depleted HUVEC treated or not with 10 μM of the ROCK1/2 inhibitor Y27632 (40 X objective, scale bar = 20 μm). (**B**) Phalloidin pixel intensity quantification normalized to the average of control HUVEC. 14 images were analyzed in each group from three independent experiments. (**C**) Central actin fiber density quantification of at least 140 cells in each group from three independent experiments. (**D**) Paracellular flux assays using confluent control and arpin-depleted HUVEC monolayers on 0.4 μm pore transwell filters treated with 10 μM Y27632 or vehicle (n=10–12 from four independent experiments). (**E**) Representative immunostaining of VE-Cadherin with F-actin staining using phalloidin in control and arpin-depleted HUVEC treated with 10 μM of the ZIPK inhibitor HS38 or vehicle (40 x objective; scale bar = 20 μm). (**F**) Phalloidin pixel intensity quantification normalized to the average of control HUVEC treated with the vehicle. 13 images were analyzed in each group from three independent experiments. (**G**) Central actin fiber density quantification of at least 125 cells in each group from three independent experiments. (**H–I**) Paracellular flux assays using confluent control and arpin-depleted HUVEC monolayers on 0.4 μm pore transwell filters treated with (**H**) 10 μM HS38 or vehicle (n=9–12 in four independent experiments) or (**I**) treated or not with 15 ng/mL TNFα and with 10 μM HS38 or vehicle (7–8 in three independent experiments). All immunofluorescences are representative of three independent experiments. All data are represented as mean ± SEM; ns: non-significant; *p<0.05; **p<0.01; ***p<0.001; ****p<0.0001; two-tailed student's t-test.

## Discussion

In this study, we discovered new and surprising arpin functions. First, arpin is expressed in endothelium in vitro and in vivo and downregulated during inflammation. Second, arpin is required for proper endothelial permeability regulation and junction architecture. Third, and most surprisingly,

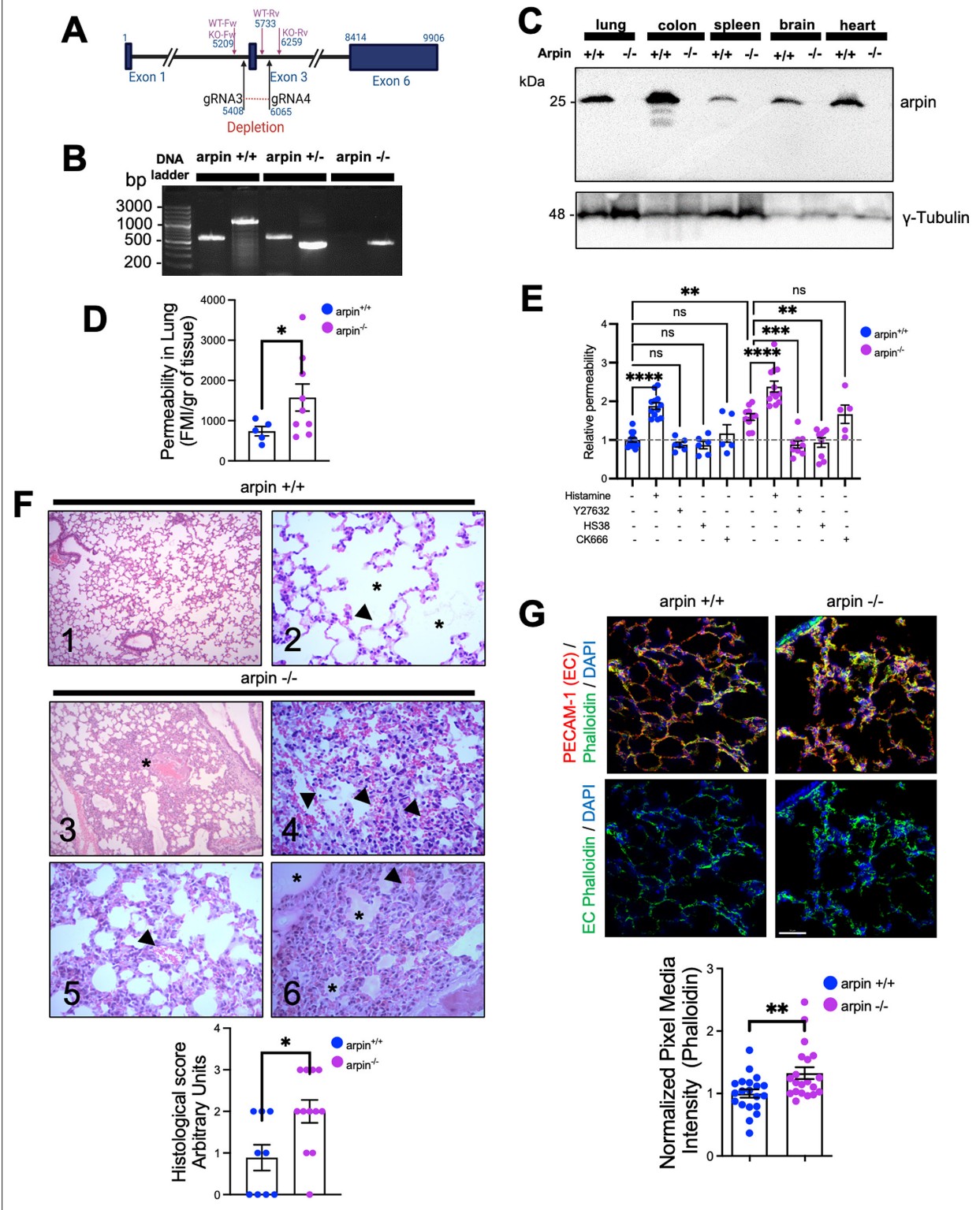

**Figure 8.** Arpin-deficient mice are viable but show increased vascular permeability in the lung and skin. (**A**) Arpin mouse gene showing exon 3, which was deleted using the CRISP/Cas9 technology to generate the complete *Arpin⁻/⁻* mouse model. (**B**) Representative genotypification PCR of mice carrying the wild-type (WT) (*Arpin⁺/⁺*), heterozygous (*Arpin⁺/⁻*), and deleted (KO, *Arpin⁻/⁻*) alleles. (**C**) Representative Western blots for arpin in lysates of the indicated organ tissues from *Arpin⁺/⁺* and *Arpin⁻/⁻* mice (n=3). (**D**) Permeability assays in the lungs. 150 kDa FITC-dextran was injected into *Arpin⁺/⁺* and *Arpin⁻/⁻* mice via the cannulated artery carotid. Animals were sacrificed, and perfused, and lungs were collected and homogenized in PBS. The homogenized tissue containing the leaked FITC-dextran was quantified using a fluorometer. (five *Arpin⁺/⁺* and eight *Arpin⁻/⁻* mice were analyzed). Data

*Figure 8 continued on next page*

*Figure 8 continued*

are represented as mean ± SEM; *p<0.05; two-tailed student's t-test with Welch's correction. (**E**) Miles assays to determine vascular permeability in the back skin of *Arpin*[+/+] and *Arpin*[-/-] mice under basal, inflammatory conditions with histamine, 10 µM of ROCK1/2 inhibitor Y27632, 10 µM of ZIPK inhibitor HS38 and 100 µM or Arp2/3 complex inhibitor CK-666. Data are shown as relative permeability with untreated *Arpin*[+/+] group set to 1 (n=at least of 7 mice in each group) and are represented as mean ± SEM. ns: non-significant; *p<0.05; **p<0.01; ***p<0.001; ****p<0.0001; two-tailed student's t-test. (**F**) Representative images of the hematoxylin and eosin staining of lung tissues of *Arpin*[+/+] (Images 1 and 2) and *Arpin*[-/-] (Images 3–6) mice. Image 1 shows normal histology (10 X objective). Image 2 shows the normal structure of the alveoli (*) and the blood vessels (arrowheads) without any pathology (40 X objective). Image 3 shows some areas with alveolar volume reduced (*) in *Arpin*[-/-] mice (10 X objective). Image 4 shows discrete interalveolar hemorrhages (arrowheads, 40 X objective). Image 5 shows congestion and dilatation of the capillaries (arrowheads, 40 X objective). Image 6 shows interalveolar edema (*) and microhemorrhage (arrowhead, 40 X objective). The histological score of the lung tissue is shown in the graph below. A score of 0 indicates no inflammation; 1 is low inflammation; 2 is moderate inflammation; 3 is high inflammation (9 images from *Arpin*[+/+] and 12 images from *Arpin*[-/-] mice were analyzed). Data are represented as mean ± SEM; *p<0.05; two-tailed student's t-test. (**G**) Representative immunostaining for PECAM-1 and F-actin staining using phalloidin in cryosections from lungs of *Arpin*[+/+] and *Arpin*[-/-] mice. Images in the bottom show endothelial cell (EC) F-actin extracted using PECAM-1 as a templete of ECs using Imaris software (40 x objective, scale bar = 50 µm). Graph shows mean F-actin pixel intensity quantification in arpin[-/-] lungs normalized to the average of images from *Arpin*[+/+] lungs (20 images were analyzed from three mice in each group). Data are represented as mean ± SEM; **p<0.01; two-tailed student's t-test.

The online version of this article includes the following source data and figure supplement(s) for figure 8:

**Source data 1.** Uncropped and labeled membranes for *Figure 8*.

**Source data 2.** Raw unedited membranes for *Figure 8*.

**Figure supplement 1.** Arpin deficiency increases vascular permeability in the skin and alters venular junction architecture in the cremaster muscle.

**Figure supplement 1—source data 1.** Uncropped and labeled membranes for *Figure 8—figure supplement 1*.

**Figure supplement 1—source data 2.** Raw unedited membranes for *Figure 8—figure supplement 1*.

---

arpin exerts these functions in an Arp2/3-independent manner; instead, it rather controls actomyosin contractility in a ROCK1/ZIPK-dependent mechanism (*Figure 9*). Finally, we present the first total arpin knock-out mice that are viable and breed and develop normally but show histological characteristics of a vascular phenotype in the lung and increased permeability similar to what we observed in arpin-depleted HUVEC.

Arpin is expressed in EC with a localization throughout the cytosol but with an enrichment at cell-cell contacts similar to what has been described in epithelial cells of the colon, and other barrier-forming cells . Such ubiquitous but diffuse subcellular location in (endothelial) cells is not surprising for an actin-regulating protein as it has been also described for other actin cytoskeleton regulators such as WAVE2 (*Mooren et al., 2014*) and the Arp2/3 complex (*Abu Taha et al., 2014*). The enrichment at junctions, however, suggested that arpin plays a role in the regulation of endothelial barrier integrity. In fibroblasts, arpin was enriched at the lamellipodial tip and accordingly regulated migration speed and directionality, one of the main functions thus far described for arpin. It will be important to reveal in future studies whether arpin also regulates endothelial cell movement for example during angiogenesis.

Of note, arpin is downregulated by pro-inflammatory cytokines suggesting that the removal of arpin is required for the changes in EC occurring during inflammation including actin cytoskeleton remodeling and junction weakening that allow for inflammation-related vascular events such as plasma protein leakage and leukocyte recruitment essential for combating the cause of inflammation and inflammation resolution. This finding is in line with our previous report showing downregulation during epithelial inflammation. Particularly, we demonstrated that arpin is not only reduced in colon epithelial cells after treatment with pro-inflammatory cytokines, but also in a DSS-induced colitis mouse model and in patients with ulcerative colitis, where loss of arpin was particularly evident in acutely inflamed tissue regions. Importantly, loss of arpin significantly correlated with epithelial barrier disruption in all models (*Chánez-Paredes et al., 2021*) indicating that loss of arpin in the context of epithelium is related to pathogenic events. This is in contrast to our new findings in endothelium where barrier weakening is required for proper inflammation resolution. However, the recovery of arpin expression during inflammation resolution and the role of arpin in vascular diseases remain to be investigated. Besides, in macrophages infected with human rhinovirus, arpin was also reduced leading to impaired phagocytosis (*Jubrail et al., 2020*). Thus, arpin downregulation seems to be relevant in different cell types to regulate different cellular functions. Our qRT-PCR results suggest that arpin reduction occurs at the transcriptional level. However, more research is needed to characterize the promoter region

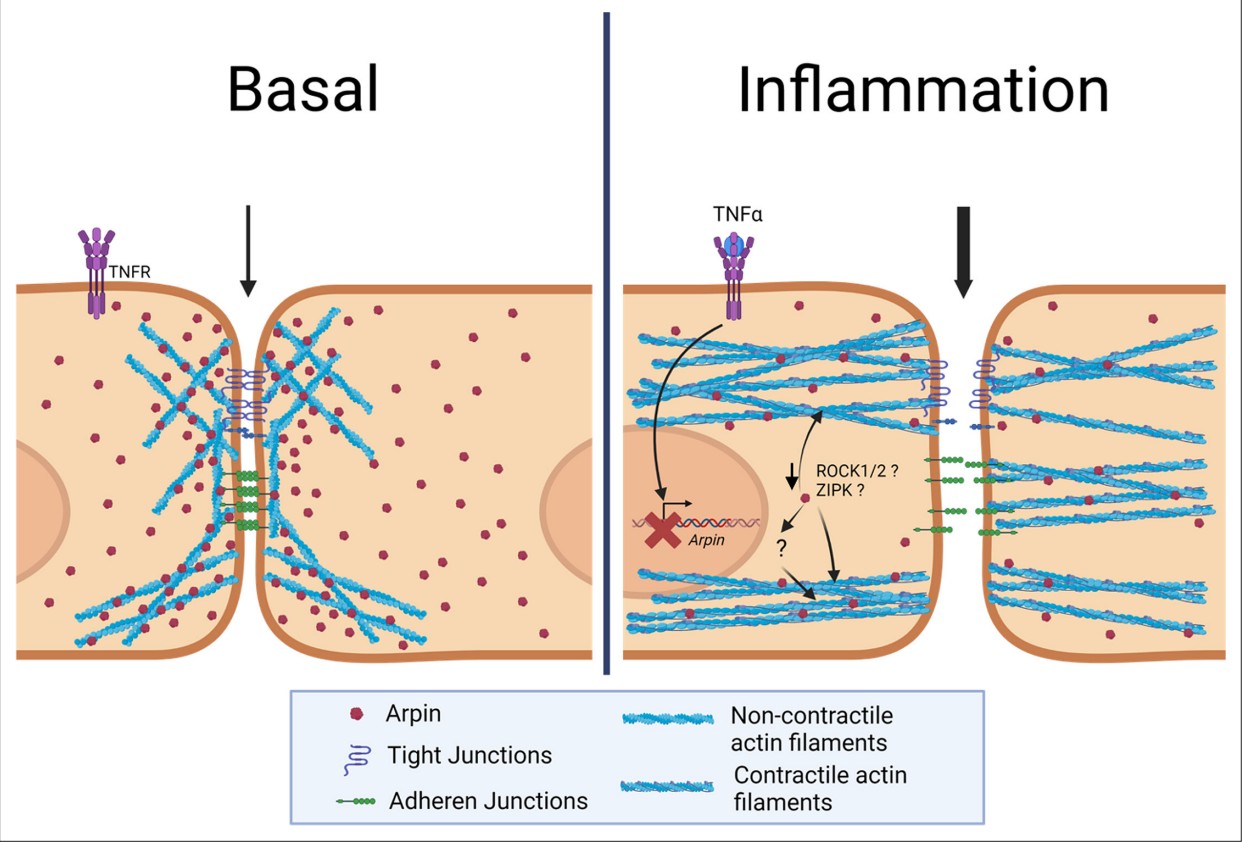

**Figure 9.** Current working model for arpin functions in endothelial cells. Under basal conditions, arpin is located throughout the cell and enriched at cellular junctions. During inflammation, arpin is downregulated causing the formation of actomyosin actin stress fibers, junction disruption and increased permeability. Question marks indicate the hitherto unknown mechanism of how arpin controls the activity of Rho-associated protein kinase (ROCK) and Zipper interacting protein kinase (ZIPK) to induce the formation of contractile actin stress fibers.

of arpin and regulatory mechanisms controlling arpin mRNA levels. Given the strong downregulation induced by cytokines such as TNFα, it is tempting to speculate that arpin mRNA levels are controlled by inflammatory pathways that are well known to be induced by TNFα such as NF-κB.

The Arp2/3 complex is critical for endothelial barrier formation, maintenance, and recovery (*Abu Taha et al., 2014*; *Belvitch et al., 2017*). Therefore, it was logical to assume that an Arp2/3 inhibitory protein such as arpin would be important for endothelial barrier integrity regulation. This idea was further supported by our study showing that arpin is involved in epithelial barrier regulation in an Arp2/3-dependent manner. Surprisingly, our results in EC suggest that arpin functions independently of the Arp2/3 complex. Instead, we described here that arpin is involved in the formation of actomyosin stress fibers in a ROCK/ZIPK-dependent fashion. It is well known that in EC ROCK1 and ZIPK are activated downstream of several pro-inflammatory mediators including TNFα (*Mong and Wang, 2009*; *Zeng et al., 2021*), which leads to the phosphorylation of MLC and MYPT1 that together drive stress fiber formation, actomyosin contractility, reduction of the barrier-stabilizing cortical actin ring and barrier dysfunction (*Millán et al., 2010*; *Angulo-Urarte et al., 2018*). Thus, our findings in arpin-depleted HUVEC of increased actomyosin stress fibers may well explain the observed increased permeability. ZIPK is not just activated but also overexpressed by TNFα in ECs (*Zeng et al., 2021*), its depletion and inhibition reduces thrombin-induced MLC phosphorylation and actomyosin contractility (*Zhang et al., 2019b*), which is in line with our observation in arpin-depleted HUVEC. Together, these data indicate a mechanistic link between arpin and ZIPK for EC barrier regulation during inflammation, but more research is required to unravel in detail the exact nature of this link. On the other hand, the spatiotemporal interplay between non-muscle-myosin II-dependent contractility and Arp2/3 complex activity regulates the formation of dactylopodia, a lamellipodia-related membrane protrusion of invasive EC implicated in sprouting angiogenesis (*Figueiredo et al., 2021*). How such interplay

may contribute to inflammatory EC barrier regulation and whether arpin connects the two different actin dynamics mechanisms are questions to be answered by future research.

Loss of the actin-binding protein (ABP) cortactin, which has also been implicated in the regulation of Arp2/3-induced actin branches, has also been shown to cause increased actomyosin contractility and as a consequence increased EC hyperpermeability (*García Ponce et al., 2016*). It remains elusive why barrier functions in EC and epithelial cells are regulated by arpin via different mechanisms, but this question is the focus of ongoing studies.

Although originally not described as an ABP, we do not discard the possibility that arpin can directly regulate actin cytoskeleton remodeling because a recently published actin interactome revealed that actin can interact transiently with arpin (*Viita et al., 2019*). Thus, it will be important to unravel whether arpin interacts with G-actin, F-actin, or both; and how this direct interaction would contribute to the regulation of the actin dynamics, junction architecture, and barrier regulation in EC.

In this study, we also report for the first time a total arpin knockout C57BL/6 mouse generated by CRISPR/Cas9-mediated genomic engineering. These mice developed and grew normally as C57BL/6 mice at least until week 12 when the arpin-deficient mice were used for experiments. Also, these mice did not show any phenotype identifiable by visual inspection. Thus, arpin is not essential for embryonic, fetal and early adult development similar to what has been described for other ABP knockout mice including those deficient for PICK1 (*Steinberg et al., 2006*), gadkin (*Schachtner et al., 2015*), cortactin (*Schnoor et al., 2011*), α-actinin3 (*MacArthur et al., 2008*) and coronin-1A (*Punwani et al., 2015*). By contrast, mice deficient for the Arp2/3 subunits ArpC4 (*van der Kammen et al., 2017*) and ArpC3 (*Yae et al., 2006*) and the NPF WAVE2 (*Dahl et al., 2003*) and N-WASP (*Snapper et al., 2001*) are not viable, thus further highlighting the difference between Arp2/3 inhibitors and ABP and the Arp2/3 complex and its activators. Importantly, our permeability and histological data revealed a clear vascular phenotype in the lung and, therefore, the importance of arpin in maintaining endothelial barrier integrity in the lung in vivo, a phenotype that is in line with our observations in arpin-depleted HUVEC. Such a strong vascular phenotype in the lung can be caused by several factors including an increase in the intravascular or interstitial hydrostatic pressure, endothelial injury or disruption of tissue barriers, or lymphatic insufficiency (*Malek and Soufi, 2023*). While this phenotype in the *Arpin⁻/⁻* mice is notorious, it apparently does not compromise the life of the mice at least into early adulthood. It will be critical to characterize older mice; and to investigate whether these mice are more susceptible to inflammatory or infectious diseases of the lung that are also characterized by the loss of tissue barrier functions. Moreover, future investigations need to analyze arpin functions in other organs such as the heart and brain in which arpin is highly expressed.

In conclusion, our results suggest that arpin, besides being an Arp2/3 inhibitor is a fundamental component of the molecular machinery that controls actomyosin contractility and endothelial barrier integrity. We propose a model in which arpin is downregulated during inflammation to induce the formation of actomyosin stress fibers and vascular permeability. This study will hopefully pave the way for additional studies on the role of arpin in vascular diseases.

## Materials and methods
### Antibodies and reagents
Polyclonal antibodies against arpin were obtained by immunizing rabbits with purified full-length human arpin (described below). Antibodies against VE-Cadherin (clone F-8), ICAM-1 (clone 15.2), β-catenin (clone E-5), and species-specific peroxidase-labeled secondary antibodies were form Santa Cruz Biotechnology (Santa Cruz, CA). Antibodies against WASP, WAVE2 (clone DSC8), p-Cofilin (S3) (clone 77G2), cofilin, pMLC (Ser19), MLC, pMYPT1 (Thr696), pMYPT1 (Thr 853), MYPT1, ROCK1 (clone C8F7), and DAPK3/ZIPK were purchased from Cell Signaling Technology (Danvers, MA). Antibodies against claudin-5 (clone 4C3C2), ZO-1, Coronin1B, and Alexa-labeled secondary antibodies were from Thermo Fisher Scientific (Waltham, MA). The antibody against vinculin was from Sigma-Aldrich (St. Louis, MO). Antibodies against PECAM-1 (clone MEC 13.3) and mDia were obtained from BD Bioscience (Franklin Lakes, NJ). The antibody against MRP14 (Clone 2B10) was purchased from US Biological Life Science (Salem, MA). Antibodies against cortactin (clone 289H10) and arpC5 (Clone 232H3) were kindly provided by Dr. Klemens Rottner and Dr. Theresia Stradal (Helmholtz Centre for Infection Research, Braunschweig, Germany). The monoclonal antibody against actin was kindly provided by

Dr. José Manuel Hernández (Department of Cell Biology, Cinvestav-IPN, Mexico City, Mexico). CK666, Y27632, and ML7 were purchased from Sigma-Aldrich. HS38 was purchased from Tocris Bioscience (Bristol, UK).

## Expression and purification of arpin

The full-length human arpin sequence cloned into the pET-32a(+) expression vector was from GenScript (Piscataway, NJ). The Arpin-pET-32a(+) construct was used to transform Rosetta (DE3) *E. coli*. Over-expression of the fusion protein was induced by adding 1 mM isopropyl-D-thiogalactopyranoside (IPTG; Sigma-Aldrich; St. Louis, MO) to the transformed cell culture. The fusion protein was purified by Ni-NTA agarose affinity chromatography (Quiagen; Valencia, CA) according to the manufacturer's instructions. Protein concentration was determined by DC Assay (Bio-Rad, Hercules, CA) and purity was analyzed by 12% SDS-PAGE.

## Production of polyclonal antibodies against arpin

One rabbit was immunized with 500 µg of recombinant arpin by subcutaneous injection of an emulsified 1:1 mixture of protein and TiterMax Gold (Sigma-Aldrich; St. Louis, MO). Three more immunizations were performed 14, 21, and 35 days after the first immunization by subcutaneous injection of 140 µg of recombinant arpin each. Pre-immune and immune sera were analyzed by ELISA and Western blot. All sera were used for Western blots and immunofluorescence stainings.

## Cell culture

HUVEC were isolated from discarded human umbilical cords as described (*Crampton et al., 2007*), and cultured in Endothelial Cell Medium (ScienCell Research Laboratories, Carlsbad, CA) at 37 °C in a humid atmosphere containing 5% $CO_2$. HUVEC were used between passages 1–7. For inflammatory conditions, confluent HUVEC monolayers were treated with 15 ng/ml tumor necrosis factor (TNF)α (Peprotech, Mexico-City, Mexico) or 15 ng/ml interleukin (IL)–1β (Peprotech) for the indicated times. Human microvascular endothelial cells (HMEC-1) were cultured in MCDB-131 medium (Sigma-Aldrich) supplemented with 10% fetal bovine serum (FBS), 1 µg/mL hydrocortisone, 20 µg/mL endothelial cell growth supplement, 10 mM L-glutamine, penicillin and streptomycin (Sigma-Aldrich) at 37 °C in a humid atmosphere containing 5% $CO_2$. The mouse brain endothelial cell line bEnd.3 was cultured in DMEM medium (Sigma-Aldrich) supplemented with 10% FBS, penicillin, and streptomycin at 37 °C in a humid atmosphere containing 5% $CO_2$. Murine lung endothelial cells (MLEC) were isolated and cultured as described (*García Ponce et al., 2016*).

## Generation of arpin-depleted HUVEC

Stable arpin-depleted HUVEC were generated by lentiviral transduction using the trans-lentiviral packaging kit (Thermo Fisher Scientific, Waltham, MA) and the pLKO.1 plasmid (Addgene, Cambridge, MA) according to the manufacturer's instructions. HUVEC were transduced in passage 2 and selected using puromycin. The following shRNA sequences were used: shCtrl: CGGAGAAGTGGAGAAGCATAC and shArpin: GGAGAACTGATCGATGTATCT (*Dang et al., 2013*).

## End-point PCR and qRT-PCR

RNA was isolated from HUVEC and HMEC-1 using TRIzol (Invitrogen, Carlsbad, CA) according to the manufacturer's instructions and treated with RNase-free DNase I (Thermo Fisher Scientific). cDNA was synthetized using the RevertAid first-strand cDNA synthesis kit (Thermo Fisher Scientific), according to the manufacturer's instructions. End-point PCR was performed using the Platinum PCR Super Mix (Thermo Fisher Scientific) with 0.15 µM of each primer and 100 µg cDNA. PCR conditions were: 95 °C for 3 min, followed by 35 cycles of 95 °C for 30 s, 55 °C for 30 s, 72 °C for 30 s, and a final extension at 72 °C for 10 min. PCR products were separated on 2% agarose gels by electrophoresis.

qRT-PCRs were carried out in a final volume of 10 µl, containing 5.0 µl Power SYBR Green PCR Master Mix (Applied Biosystems, Foster City, CA), 0.15 µM of each primer and 125 µg cDNA using a StepOne Real-Time PCR System (Applied Biosystems). PCR conditions were: 95 °C for 10 min, followed by 40 cycles of 95 °C for 15, 60 °C for 60 s, and the melting curve was generated during the last cycle by heating from 60–95°C in increments of 0.5 °C/s. Relative expression was quantified using the $2^{-\Delta\Delta CT}$ method with 7SL as the housekeeping gene (*Rao et al., 2013*). Primer sequences were:

*7SL*-Fw: ATCGGGTGTCCGCACTAAGTT, *7SL*-Rv: CAGCACGGGAGTTTTGACCT, *ARPIN*-Fw: GCCC AGAGTCACACAGCTAA, *ARPIN*-Rv: CTTTCTGAAGGGCAAGGAAG, *PICK1*-Fw: ATGATTCAGGAG GTGAAGGG, *PICK1*-Rv: CGGTGCTTGACTTTCTTCAA, *AP1AR*-Fw: GAAATGACGACAGCACATCC, *AP1AR*-Rv: TAAGTGCTGCCCGTAGAATG, *Arpin*-Fw: GCTCTCTGTCAATTCCAGCA, *Arpin*-Rv: CACC AGGAAGGTGAACACAG, *Actb*-Fw: TATCCACCTTCCAGCAGATGT, *Actb*-Rv: AGCTCAGTAACA GTCCGCCTA.

## Western blot

Equal amounts of protein lysates from cells or tissues were separated by SDS-PAGE, transferred onto 0.45 μm pore nitrocellulose membranes (Bio-Rad, Hercules, CA), and then blocked with Tris-buffered saline (TBS) containing 0.05% Tween and 5% skim-milk or 5% bovine serum albumin (BSA) for 1 hr at RT. Immunoblots were incubated with primary antibodies overnight at 4 °C, washed, and then incubated with HRP-conjugated secondary antibodies for 1 hr at RT. Protein bands were visualized and recorded using a ChemiDoc imaging device (Bio-Rad). Pixel intensities of the bands were quantified using ImageJ software (NIH, Bethesda, MD).

## Paracellular flux assay

60,000 HUVECs were seeded on 6.5-mm-diameter 0.4 μm pore transwell filters Corning-Costar, Acton, MA, coated with 0.8% gelatin and cultivated for 72 hr. Then, 50 μg of 150 kDa FITC-dextran (Sigma-Aldrich) were added to the upper chamber with fresh medium in the lower chamber and incubated for 1 hr at 37 °C. Then, 100 μL medium was taken from the lower chamber and the fluorescence was quantified using a fluorometer. Emission values were normalized to control cells (set to 1).

## Calcium-switch assay

HUVEC were cultivated on coverslips coated with 0.8% gelatin until confluent. To generate a calcium-free environment, the cells were cultivated for 2 hr in ECM medium in the presence of 2.5 mM EGTA to chelate-free calcium. After 2 hr, the medium was changed to a normal ECM medium with calcium. Cells were cultivated for the indicated times, then fixed and stained for VE-Cadherin as described below.

## Immunofluorescence staining and microscopy

HUVEC were seeded on glass coverslips coated with 0.8% gelatin and cultivated to confluence. HUVEC were fixed with 4% paraformaldehyde (PFA) for 10 min at RT and then permeabilized with 0.1% Triton X-100 in PBS for 10 min at 4 °C. Then, samples were blocked for 30 min with 3% BSA in PBS. Incubation with primary antibodies in 3% BSA in PBS was performed overnight at 4 °C followed by washing and incubation with phalloidin labeled with Alexa-Flour 488. Biolegend, San Diego, CA and species-specific fluorescently labeled secondary antibodies for 1 hr at RT. Preparations were mounted in Vecta-Shield containing 4'6-diamidino-2-phenylindole (DAPI; Thermo Fisher Scientific).

For lung tissue staining, 8–12 weeks-old male mice were anesthetized and sacrificed. Then a 1:1 dilution of Tissue-Tek (Fisher Health Care, Waltham, MA) and PBS was injected via the trachea into the lungs, and the trachea was ligated to avoid leakage. The lungs were extracted and embedded in Tissue-Tek. The samples were frozen, cut into 8 μm sections using a cryostat (Leica, Wetzlar, Germany), and collected on glass slides. Subsequently, tissue sections were fixed in absolute ethanol at –20 °C for 30 min and incubated with 0.2% NaHB$_4$ for 5 min at RT to reduce autofluorescence. Samples were blocked with 6% BSA in PBS + 0.01% Tween. Incubation with primary antibodies in 6% BSA in PBS was performed overnight at 4 °C followed by washing and incubation with phalloidin labeled with Rhodamine (Thermo Fisher Scientific) and species-specific fluorescently labeled secondary antibodies for 1 hr at RT. The samples were incubated in 0.1% Sudan Black B diluted in 70% ethanol for 5 min at RT to further reduce autofluorescence. Finally, preparations were washed with water and mounted in Vecta-Shield containing DAPI.

For cremaster tissue staining, 8–12 weeks old male mice were anesthetized, and sacrificed, and the cremasters collected and fixed in 4% PFA for 45 min at 4 °C. Next, the tissues were blocked and permeabilized with 25% FBS, and 0.5% Triton X-100 in PBS for 4 hr at RT. Incubation with primary antibodies in 10% FBS in PBS was performed overnight at 4 °C followed by washing and incubation with species-specific fluorescently labeled secondary antibodies for 2 hr at RT. Finally, cremaster tissues

were washed with PBS and mounted. All samples were examined using a laser confocal microscope (Leica TCS SP8).

## Generation of arpin-deficient mice

The *Arpin*⁻/⁻ mice on a C57Bl/6 genetic background were generated by CYAGEN (Santa Clara, CA) using the CRISPR/Cas9 technology with two gRNA sequences (gRNA3: CTATGCAGCAGGGTAG CGCCAGG and gRNA4: CGCACAGTGATAGTCGGAGCCGG) that target the exon 3 of the arpin mouse gene (*Figure 3A*). Cas9 mRNA and gRNA sequences were co-injected into fertilized mouse eggs to generate targeted knockout offspring. F0 founder animals were identified by genotyping and sequencing. Positive founders were bred with WT mice to test germline transmission. Positive F1 animals were shipped and used to establish a colony at the animal facility of Cinvestav-IPN. All animal experiments were approved by the institutional animal care and use committee and the bioethics committee of Cinvestav with protocol number 0227–16.

## Genotyping

Genomic DNA was prepared from mouse tails dissolved in 12 mM NaOH and 0.2 mM EDTA and incubated at 98 °C for 1 hr, followed by the addition of 40 mM Tris-HCl pH 5.5. Genotyping to detect the WT and arpin-targeted alleles were performed in two different PCRs using the following primers: forward GCTGGCAACTTCAATCCTGCCT; and reverse GAGACAAACAAGTCAGTCTAACAGCCTC, producing a PCR fragment of 552 bp derived from the WT allele only; and forward GCTGGCAACTTC AATCCTGCCT; and reverse CAGTGTTCCCAGGGCTTGTCTGA, producing PCR fragments with sizes of 1073 bp (WT allele) and ~420 bp (mutant KO allele) (*Figure 3B*). PCR was performed in a total volume of 10 μL and PCR conditions were: 94 °C for 5 min, followed by 35 cycles of 94 °C for 30 s, 60 °C for 30 s, and 72 °C for 60 s, and a final extension at 72 °C for 5 min. The PCR products were separated by electrophoresis on 2% agarose gels.

## Histology

Lungs were embedded in paraffin using standard protocols. Then samples were cut into 8 μm sections, mounted on glass slides and stained with hematoxylin, and eosin according to standard protocols. The histological analysis and score calculation were done by a pathologist in a blinded fashion. The histological score considers the presence or absence of edema, hemorrhage, congestion, immune cells in the tissue, and the extent of all these parameters (*Lartey et al., 2022b*; *Lartey et al., 2022a*). A score of 0 indicates no inflammation; 1 is low inflammation; 2 is moderate inflammation; 3 is high inflammation.

## Determination of vascular permeability

For permeability assays in the lungs, *Arpin*⁺/⁺ and *Arpin*⁻/⁻ male mice, between 8–12 weeks old, were anesthetized and injected with 50 mg of 150 kDa FITC-dextran (Sigma-Aldrich) per 1 kg of body weight via the cannulated carotid artery. After 30 min, the mice were perfused with PBS to remove all the blood from the circulation. Then, the lungs were collected, weighted, and homogenized in PBS. Fluorescence in the tissue was measured in a fluorometer and the mean fluorescence intensity (MFI) was normalized to tissue weight.

Modified Miles assays to measure vascular permeability in the skin was performed by injecting 8–12 weeks old *Arpin*⁺/⁺ and *Arpin*⁻/⁻ mice intraperitoneally with 2% Evans Blue dye in PBS. After 2 hr, mice were anesthetized and 100 μL of PBS containing 450 ng Histamine or 10 μM ROCK1/2 inhibitor Y27632 or 10 μM ZIPK inhibitor HS38, or 100 μM Arp2/3 complex inhibitor CK-666 or PBS only were intradermally injected into the shaved back skin. After 30 min, mice were sacrificed, and the skin around the injection sites were excised (circles of 1 cm in diameter). The skin pieces were incubated separately in formamide for 24 hr to extract the blue dye and then read spectrophotometrically at 600 nm. Tissues were dried for 8 days at RT and subsequently weighted. The densitometry values at 600 nm were divided by tissue weight and normalized to the values of the *Arpin*⁺/⁺ mice.

## Blood leukocyte counts

Whole blood was collected from anesthetized 8–12 weeks old mice by heart puncture with heparin as an anti-coagulant. Blood samples were analyzed using an Exigo Veterinary Hematology System (Stockholm, Sweden) according to the manufacturer's instructions.

## Statistics

Statistical analyses were performed using Prism 8.0 software (GraphPad). Significance between the two groups was assessed by a two-tailed Student's t-test or a two-tailed Student's t-test with Welch's correction. Two-way ANOVA was performed for multiple group comparisons. For the mating statistics, $\chi^2$ tests were performed to test the probability of the deviation between the observed and the expected values.

## Acknowledgements

This project was funded by the Mexican National Council for Science and Technology (Consejo Nacional de Ciencia y Tecnología, CONACyT, Basic Science Project 284292 to MS). We thank all veterinarians in the animal facility of Cinvestav-IPN for taking care of all the mice; and the confocal microscopy unit at Cinvestav-IPN for helping with image acquisition. We also thank Dr. Klemens Rottner and Dr. Theresia Stradal (Technical University Braunschweig and Helmholtz Center for Infection Research, Braunschweig, Germany) for providing antibodies against arpC5 and cortactin; and Dr. Manuel Hernández-Hernández, Department of Cell Biology, Cinvestav-IPN, for proving antibodies against actin.

## Additional information

### Funding

| Funder | Grant reference number | Author |
| --- | --- | --- |
| Consejo Nacional de Ciencia y Tecnología | 284292 | Michael Schnoor |

The funders had no role in study design, data collection and interpretation, or the decision to submit the work for publication.

### Author contributions

Armando Montoya-Garcia, Formal analysis, Investigation, Methodology, Writing - original draft, Writing - review and editing; Idaira M Guerrero-Fonseca, Abigail Betanzos, Ricardo Mondragon-Flores, Formal analysis, Investigation, Methodology; Sandra D Chanez-Paredes, Karina B Hernandez-Almaraz, Iliana I Leon-Vega, Citlaltepetl Salinas-Lara, Formal analysis, Investigation; Angelica Silva-Olivares, Hilda Vargas-Robles, Investigation; Monica Mondragon-Castelan, Investigation, Methodology; Michael Schnoor, Conceptualization, Supervision, Funding acquisition, Methodology, Writing - original draft, Writing - review and editing

### Author ORCIDs

Armando Montoya-Garcia (ID) http://orcid.org/0009-0008-0590-8292
Idaira M Guerrero-Fonseca (ID) https://orcid.org/0000-0003-0190-7571
Abigail Betanzos (ID) https://orcid.org/0000-0003-1761-0481
Michael Schnoor (ID) https://orcid.org/0000-0002-0269-5884

### Ethics

This study was performed in strict accordance with the Mexican laws. All of the animals were handled according to approved institutional animal care and use committee (IACUC) protocols (protocol 0227-16) of Cinvestav-IPN. All experiments were performed under anesthesia, and every effort was made to minimize suffering.

Reviewer #1 (Public review): https://doi.org/10.7554/eLife.90692.3.sa1

Reviewer #2 (Public review): https://doi.org/10.7554/eLife.90692.3.sa2
Author response https://doi.org/10.7554/eLife.90692.3.sa3

## Additional files

### Supplementary files

• Supplementary file 1. Table showing the mating statistics of *Arpin⁺/⁻* mice. Genotyping of 126 littermates from breedings of heterozygous (*Arpin⁺/⁻*) mice are shown. $\chi$2 test probability is shown. Deviations from Mendelian rules are non-significant.

• Supplementary file 2. Hemograms of *Arpin⁺/⁺* and *Arpin⁻/⁻* mice.

• MDAR checklist

### Data availability

All data generated or analysed during this study are included in the manuscript and supporting files; source data files have been provided for Figures 1-4, 6, 8, Figure 2—figure supplement 1, and Figure 8—figure supplement 1.

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
