## [Editor Report · eLife assessment]

This study presents **solid** results to demonstrate that arpin is expressed in the endothelium of blood vessels and that its deficiency leads to leaky blood vessels in in vivo and in vitro models. The work does not yet clarify the mechanistic connection between arpin and increased ROCK activity. The study adds some insights to our understanding of the complicated network of proteins that control this process, and it will be **useful** to individuals within this defined field of study.

---

## [Referee Report · Reviewer #1 (Public review)]

Summary:

The data clearly demonstrate that arpin is important for vessel barrier function, yet its genetic loss via a CRISPR strategy was not lethality, but led to viable animals in C57Blk strain at 12 weeks of age, albeit with leaky blood vessels. Pharmacological approaches were employed to demonstrate that loss of arpin led to ROCK1-dependent stress fiber formation that promoted increased permeability.

Strengths:

The results clearly demonstrate that arpin is expressed in the endothelium of blood vessels and its deficiency leads to leaky blood vessels in in vivo and in vitro models.

Weaknesses:

They conclude vessel leak was not related to enhanced Arp2/3 function through arpin deficiency, but no direct evidence of Arp2/3 activity is provided to support this conclusion. Instead, the authors concluded that ROCK1 activity was elevated in arpin knockdown cells and caused robust stress fiber formation. This idea could be strengthened by testing if ROCK1 inhibition by pharmacological block in arpin KO mice leads to less vascular leakage while pharmacological inhibition of Arp2/3 does not attenuate increased vessel permeability.

---

## [Referee Report · Reviewer #2 (Public review)]

Summary:

The authors have taken their previous finding that arpin is important for epithelial junctions and extended this to endothelial cells. They find that the positive effects of arpin on endothelial junctions are not dependent on Arp2/3 activity but instead on suppression of actinomyosin contractility.

Strengths:

The study uses standard approaches to test each of the components in the model. The quality of the experimental work is good and the amount of experimental evidence is sufficient to support this straightforward story.

Weaknesses:

The major weakness is that the story is a simple extension of the previous work on arpin and junctions in epithelial cells. The additional information is that the effects are not via Arp2/3 directly, but instead through an increase in actinomyosin contractility. However, the connection between arpin and increased ROCK activity is not revealed.

---

## [Author Response]

The following is the authors’ response to the original reviews.

**Recommendations for the authors:**

**Reviewer #1 (Recommendations for the Authors):**
Arpin is a negative regulator of Arp2/3 activity. Here the authors investigated the role of arpin in vascular permeability using appropriate cultured human and murine endothelial monolayers and successfully developed an arpin KO mice. The results clearly show arpin is expressed in blood vessels (not clear about lymphatics but given leaky vessels, one wonders). The data show that arpin is important for vessel barrier function yet its genetic loss still leads to viable animals in the C57Blk strain albeit with leaky blood vessels. The data are well presented and controls are included. However, the evidence that arpin loss/knockdown causes increased actin functions independent of Arp2/3 is based on pharmacological data and is indirect. Authors conclude ROCK1 activity is elevated and the cause of lost barrier function by arpin reduction. I do have one suggestion for the authors that involves a new study in these animals, which could strengthen their proposed mechanism that the vascular defects are independent of Arp2/3 activity and rather involve ROCK1 but not ZIPK.(1) If arpin is working via ROCK1, as the authors infer, perhaps treatment of arpin-/- mice with ROCK1 inhibitor(s) would attenuate vessel permeability while HS38 treatment would not. This type of study would strengthen the conclusion that ROCK1, but not ZIPK, was involved. Including CK666 if active in mouse cells, could also be tested.

To analyze vascular permeability in vivo, we performed Miles assays in arpin+/+ and arpin-/- mice using the inhibitors of ROCK1 (Y27632) and ZIPK (HS38). Both Y27632 and HS38 reduced the permeability caused by absence of arpin (new Figure 8E), thus confirming what we observed before in HUVEC (shown in old Figure 7). CK666 did not change the permeability in arpin-/- mice, thus confirming the conclusion that arpin does not regulate vascular permeability via Arp2/3 but rather via ROCK1/ZIPK-mediated stress fiber formation (page 13).

(2) Fig 5. Data demonstrate that Arpin regulates actin filament formations and permeability in HUVEC, but this does not demonstrate its occurring in an Arp2/3-independent manner. If I understand your data this is indirect evidence. One needs more information to reach this conclusion. Can authors measure Arp2/3 directly and then test whether arpin knockdown and CK666 have the same capacity to reduce Arp2/3 activity in vitro.

Arp2/3 activity cannot be measured directly. The commonly used approach is therefore Arp2/3 inhibition via CK666. Our new in vivo permeability assays (see answer above) together with our HUVEC data in Figure 5 clearly show that CK666 does not have the same effect as arpin knock-down, and neither does CK666 rescue the effects of arpin deficiency in vitro and in vivo. Together, these findings clearly suggest that arpin does not regulate endothelial permeability via Arp2/3.

Minor issues:Fig 2, 3 or other Figs: In presented western blots, all proteins should include appropriate mw labels.

Thank you. Molecular weights have been added to all Western blots.

Fig 2. Suggest that like your arpin analysis, amounts of AP1AP and PICK1 at baseline and TNF-treatment by blotting should be included. A minor point is yellow color for labels does not stand out and should be changed to another color - as the authors used in Fig 2C.

We have included Western blots and quantifications for PICK1 in Figure S1A and S1C. An antibody against AP1AP was unfortunately not available.

The yellow color has been changed to purple for better visibility.

Fig 2C. The arpin loss at junctions and actin filaments (Figure 2C) is very minor even though it reached statistical significance. It really is not an obvious loss from your 3 color overlay.

Thank you. It is indeed hard to see. We included now magnifications in Figure 2C that better show the loss of arpin at junctions.

Fig 8, text 303-310 shows in vivo evidence of lung congestion and edema. Also appear to be inflammatory cells present in images. If these are inflammatory cells, it begs the question if these mice have an abnormal complete blood cell count (CBC). Suggest adding CBC data for arpin-/- vs control arpin +/+ mice in Fig 8.

The pathologist observed the presence of lymphocytes and macrophages, indicating the possibility of a (low level) chronic inflammation in arpin-deficient lungs. However, we now also performed hemograms of the mice (new Table S2) that showed no significant difference in the blood cell count of arpin-/- and arpin+/+ mice. Thus, the presence of lymphocytes and macrophages cannot be explained simply by higher leukocyte counts (page 14).

Line 289, pg 13, Fig 8: Lung levels of arpin are not shown in Fig 8B. Authors must mean another fig?

Sorry. Arpin protein levels in lungs are shown in figure 8C. This has been corrected on page 13.

**Reviewer #2 (Recommendations For The Authors):**
This is a solid piece of work that adds a small amount of additional factual information to our understanding of cell-cell junctions. The experimental work is of good quality and is sufficient to support the aims of the paper. I think the value of the work is to add this small amount of new knowledge to the archive. I do not believe that further experimental work would add to the paper - it's done. But this doesn't have the impact or completeness for this journal. It belongs in a for-the-record journal.

We appreciate your overall positive evaluation and your comments that our study represents a solid piece of work with good quality experimental work. However, we are not sure what you mean by “it belongs in a for-the-record journal”. Anyway, we agree that our study does not reveal a complete mechanism of how arpin regulates actin stress fibers, but we respectfully disagree that our study only adds a “small amount of additional factual information”. We may not have been very clear about it, but we present in this study several new discoveries and although some are descriptive in nature that does not make them trivial or less important. We provide for the first time experimental evidence that: (1) arpin is expressed in endothelial cells in vitro and in vivo, and downregulated during inflammation; (2) presence of arpin is required for proper endothelial permeability regulation and junction architecture; (3) arpin exerts these functions in an Arp2/3-independent manner; (4) arpin controls actomyosin contractility in a ROCK1- and ZIPK-dependent fashion; (5) arpin knock-out mice are viable and breed and develop normally but show histological characteristics of a vascular phenotype and increased vascular permeability that can be rescued by inhibition of ROCK1 and ZIPK. The fact that arpin fulfills its functions in endothelial cells independently of the Arp2/3 complex is of special relevance as previously the only known function of arpin was the inhibition of the Arp2/3 complex. Thus, we believe that our study adds a significant amount of new information to the literature. Thank you very much.